# EFFICIENT EDGE INFERENCE BY SELECTIVE QUERY

**Anil Kag**
Boston University
anilkag@bu.edu

**Igor Fedorov**
Meta AI
ifedorov@meta.com

**Aditya Gangrade**
Carnegie Mellon University
agangra2@andrew.cmu.edu

**Paul Whatmough**[*]
Qualcomm AI Research
pwhatmou@qti.qualcomm.com

**Venkatesh Saligrama**
Boston University
srv@bu.edu

## ABSTRACT

Edge devices provide inference on predictive tasks to many end-users. However, deploying deep neural networks that achieve state-of-the-art accuracy on these devices is infeasible due to edge resource constraints. Nevertheless, cloud-only processing, the de-facto standard, is also problematic, since uploading large amounts of data imposes severe communication bottlenecks. We propose a novel end-to-end hybrid learning framework that allows the edge to selectively query only those hard examples that the cloud can classify correctly. Our framework optimizes over neural architectures and trains edge predictors and routing models so that the overall accuracy remains high while minimizing the overall latency. Training a hybrid learner is difficult since we lack annotations of hard edge-examples. We introduce a novel proxy supervision in this context and show that our method adapts seamlessly and near optimally across different latency regimes. On the ImageNet dataset, our proposed method deployed on a micro-controller unit exhibits 25% reduction in latency compared to cloud-only processing while suffering no excess loss.

## 1 INTRODUCTION

We are in the midst of a mobile and wearable technology revolution with users interacting with personal assistants through speech and image interfaces (Alexa, Apple Siri etc.). To ensure an accurate response, the current industrial practice has been to transmit user queries to the cloud-server, where it can be processed by powerful Deep Neural Networks (DNNs). This is beginning to change (see (Kang et al., 2017; Kumar et al., 2020)) with the advent of high-dimensional speech or image inputs. As this interface gains more traction among users, cloud-side processing encumbers higher latencies due to communication and server bottlenecks. Prior works propose a hybrid system whereby the edge and cloud-server share processing to optimize average latency without degrading accuracy.

**Proposed Hybrid Learning Method.** Our paper focuses on learning aspects of the hybrid system. We propose an end-to-end framework to systematically train hybrid models to optimize average latency under an allowable accuracy-degradation constraint. When a user presents a query, the hybrid learner (see Fig. 1) decides whether it can respond to it on-device (eg. "Can you recognize me?") or that the query posed is difficult (eg. "Play a song I would like from 50's"), and needs deeper cloud-processing. We emphasize that due to the unpredictable nature (difficulty and timing) of queries coupled with the fact that on-device storage/run-time footprint is relatively small, hard queries inevitably encumber large latencies as they must be transmitted to the cloud.

*Fundamental Learning Problem: What queries to cover?* While, at a systems level, communications and device hardware are improving, the overall goal of maximizing on-device processing across users (as a way to reduce server/communication loads) in light of unpredictable queries is unlikely to change. It leads to a fundamental learning problem faced by the hybrid learner. Namely, how to train a base, a router, and the cloud model such that, on average, coverage on-device is maximized without sacrificing accuracy. In this context, coverage refers to the fraction of queries inferred by the base model.

---

[*]Work completed while PW was at Arm Research

**Figure 1:** HYBRID MODEL. Cheap base ($b$) & routing models ($r$) run on a micro-controller; Expensive global model ($g$) runs on a cloud. $r$ uses $x$ and features of $b$ to decide if $g$ is evaluated or not.

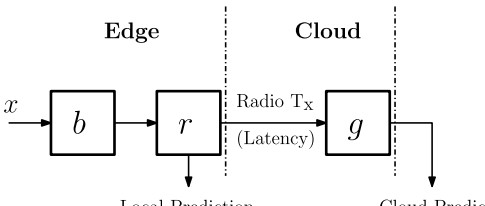

**Table 1:** Device & Model Characteristics: Edge (STM32F746 MCU), Cloud (V100 GPU). It takes 2000ms to communicate an ImageNet image from the edge to the cloud (see Appendix §A.2)

| Device | Device Characteristics | | Model Performance | | |
|--------|--------|--------|--------|--------|--------|
| | Memory | Storage | Accuracy | Size | Latency |
| MCU | 320KB | 1MB | 51.1% | 0.6MB | 200ms |
| GPU | 16GB | 1TB | 79.9% | 9.1MB | 25ms |

**Novel Proxy Supervision.** Training routing models is difficult because we do not a priori know what examples are hard to classify on edge. More importantly, we only benefit from transmitting those hard-to-learn examples that the cloud model correctly predicts. In this context, we encounter three situations, and depending on the operational regime of the hybrid system, different strategies may be required. To expose these issues, consider an instance-label pair $(x, y)$, and the three typical possibilities that arise are: (a) Edge and Cloud are both accurate, $b(x) = g(x) = y$; (b) Edge and Cloud are both inaccurate, $b(x) \neq y$ and $g(x) \neq y$; and (c) Edge is inaccurate but Cloud is accurate $b(x) \neq g(x) = y$. Our objective is to transmit only those examples satisfying the last condition. *It improves coverage by ceasing data transfers when cloud predictions are bound to be incorrect.* However, the limited capacity of the router (deployed on the MCU) limits how well one can discern which of these cases are true, and generalization to the test dataset is difficult. As such, the routing would benefit from supervision, and we introduce a novel *proxy supervision* (see Sec. 2.1) to learn routing models while accounting for the base and global predictions.

*Latency.* Coverage has a one-to-one correspondence with average latency, and as such our method can be adopted to maximizes accuracy for any level of coverage (latency), and in doing so we characterize the entire frontier of coverage-accuracy trade-off.

**Contributions.** In summary, we list our contributions below.

- *Novel End-to-End Objective.* We are the first to propose a novel global objective for hybrid learning that systematically learns all of the components, base, router, global model, and architectures under overall target error or dynamic target latency constraints.
- *Proxy Supervision.* We are the first to provably reduce router learning to binary classification and exploit it for end-to-end training based on novel proxy supervision of routing models. Our method adapts seamlessly and near optimally across different latency regimes (see knee in Fig. 2).
- *Hardware Agnostic.* Our method is hardware agnostic and generalizes to any edge device (ranging from micro-controllers to mobile phones), any server/cloud, and any communication scenarios. Our experiments include (a) MCU and GPU (see Sec. 3.1), (b) Mobile Devices and GPUs (see Sec. 3.1), (c) on the same device (see Sec. 3.2, Appendix A.11.1).
- *SOTA Performance on Benchmark Datasets.* We run extensive experiments on benchmark datasets to show that the hybrid design reduces inference latency as well as energy consumption per inference. Our code is available at `https://github.com/anilkagak2/Hybrid_Models`

**Motivating Example: ImageNet Classification on an MCU.** Let us examine a large-scale classification task through the lens of a tiny edge device. This scenario will highlight our proposed on-device coverage maximization problem. We emphasize that the methods proposed in this paper generalize to any hardware and latencies (see Section 3). Table 1 displays accuracy and processing latencies on a typical edge (MCU) model and a cloud model (see Appendix §A.2). The cloud has GPUs, and its processing speed is 10x relative to MCU processing. In addition, the cloud has a much larger working memory and model storage space compared to the MCU.

*Impact of Cloud-Side Bottlenecks.* If latency were negligible (either due to communication speed or server occupancy), we would simply transfer every query to the cloud. In a typical system with NB-IoT communication (see Sec. A.2), data transfer to the cloud can take about $2s$—about $100\times$ the processing time on a GPU—and thus, to reduce communication, one should maximize MCU utilization for classification. The argument from an energy utilization viewpoint is similar; it costs $20\times$ more for transmitting than processing. In general, communication latency is dynamically changing over time. For instance, it could be as large as 10x the typical rate. In this case, we would want the

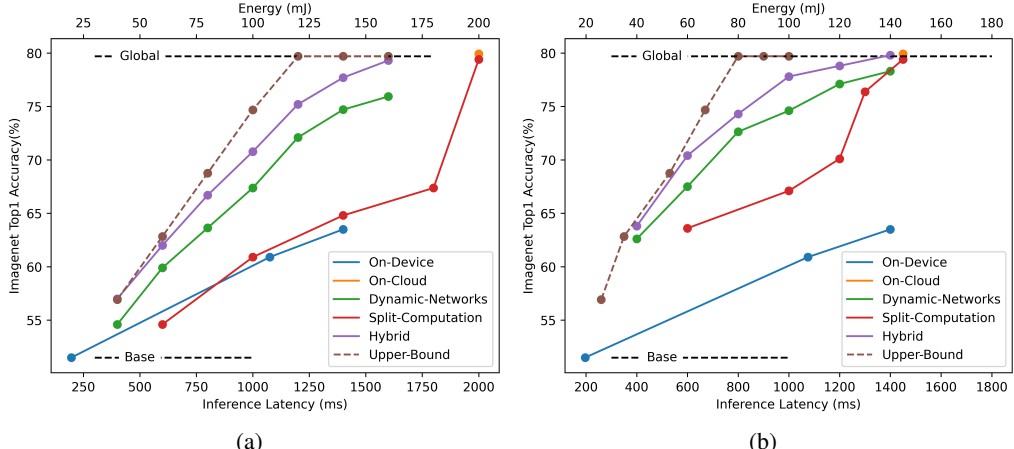

**Figure 2:** ImagetNet Classification: Accuracy vs Energy/Latency plot: (a) Constant Communication Latency (2000ms), and (b) Dynamic Communication latency [200, 2000]ms. It clearly shows that the proposed hybrid model pareto dominates the baselines while getting significantly closer to the upper-bound in hybrid setup.

MCU unit to dynamically adapt its processing so that at fast rates, much of the processing happens on the cloud, and for slow rates, the MCU must still decide what examples to process locally.

## 1.1 Prior Works with Empirical Comparisons on MCU

We use Table 1 as a running example to survey prior works, but our inferences here, as seen in Sec. 3 hold generally. Our goal is to illustrate key differences through this practical scenario. We refer the reader to Appendix A.2 for more experimental details. We consider two communication contexts: (a) Fixed but large latency (2000ms); (b) Dynamic Latency: A random variable uniformly distributed in the interval [200, 2000]ms.

**Table 2:** Comparing features of our proposal against baseline. E.-to-E. stands for 'End-to-End', and Arch. for 'Architecture'.

| Method | Low Latency | Deploy. on Edge | E.-to-E. Training | Arch. Search | High Accuracy |
|---|---|---|---|---|---|
| On-Device | ✓ | ✓ | - | ✓ | ✗ |
| On-Cloud | ✗ | ✗ | - | ✓ | ✓ |
| Split-Comp. | ✗ | ✗ | ✓ | ✗ | ✗ |
| Dynamic. | ✓ | ✓ | ✗ | ✗ | ✗ |
| Hybrid | ✓ | ✓ | ✓ | ✓ | ✓ |

**All-Cloud Solution.** Here, the edge device transmits all examples to the cloud. It incurs a communication cost in addition to the cloud inference latency. This solution is sub-optimal as seen by the orange circle with 80% accuracy and inference latency of 2025ms in Figure 2(a). Our goals are:
(i) To understand the trade-off between latency and accuracy;
(ii) To compare prior methods by the latency level at which cloud-accuracy is achieved.

**All-Edge Solution.** While we can leverage various methods (Howard et al., 2019; Hinton et al., 2015; Cai et al., 2020; Kag & Saligrama, 2022), MCUnet Lin et al. (2020) is the most suitable deployable model for this micro-controller. [1] All-edge baseline is blue curve in Figure 2(a). The smallest MCUNet model achieves 51.1% accuracy with 200ms inference latency. In contrast, the largest MCUNet model achieves 63.5% accuracy with 1400ms inference latency; models any larger cannot be deployed due to the MCU hardware limitations listed in Table 1. Thus, *cloud-accuracy is unachievable with an all-edge solution for any latency.*

**Split-Computation methods.** Kang et al. (2017) splits the model between the edge and cloud device. More specifically, the initial part of the DNN is stored and executed on the edge device. Then, the edge transmits the intermediate features to the cloud, where the remaining computation of the DNN is performed. This split takes into account the latency budget and the edge transmission cost. More recent variants (Li et al., 2021; Odema et al., 2021) add a branch-out stage that allows for classification on the edge device whenever the decision is deemed correct. We employ this method in our

---

[1]Although Lin et al. (2021) improves upon MCUNet, their pre-trained models are not available publicly. Besides, any improvement in the edge model would yield improvements in all the hybrid baselines.

comparisons. A fundamental issue is that requiring copies of the same model on server/edge severely constrains practical use cases. For instance, in Table 1 server model is 9X of MCU model. In this case, we could either use the maximum-size edge model and copy it to the server or store a partial copy of the server model locally. The former leads to topping accuracy at 63.5% (blue-line) at high latency, while the latter is plotted as the red curves in Figure 2. While results improve in the dynamic setting, they are not comparable to the other baselines. This fact is not surprising since the amount of pre-processing gained by having a shared model is more than offset by allowing models optimized to the edge. Since the MCUs are too small, only a tiny part of the large DNN can be stored/executed on the edge device. The resulting intermediate feature size is larger than the input size. Thus, cloud accuracy can only be achieved by transmitting all examples to the cloud at a latency of an all-cloud solution.

**Dynamic Neural Network** methods are surveyed in (Han et al., 2021). These methods budget more compute for hard instances. It includes (a) cascade-based early exit networks (Park et al., 2015; Bolukbasi et al., 2017; Wang et al., 2018), where the constituents are independently designed without feature sharing; and (b) early exit networks (Teerapittayanon et al., 2017; Dai et al., 2020; Li et al., 2019) where classifiers are introduced at intermediate layers. These works generally disregard the severely resource-constrained setting such as in Table 1. Within this framework, we experimented with different approaches and found that (Bolukbasi et al., 2017; Nan & Saligrama, 2017; Li et al., 2021) serves as a good baseline. These allow for different models optimized for the base and global, but the routing scheme is designed post-hoc based on classifying locally depending on prediction uncertainty. This is depicted as green-line in Figure 2.

**Proposed hybrid method**, in contrast to Dynamic Networks, globally optimizes all of the parts (base, router, and cloud) and is the purple line in Figure 2. It is evidently close to the upper bound, which is derived in Appendix A.2 based on the practical assumption that the base model is agnostic to what examples are classified by cloud correctly. Prior works including dynamic networks and split-computation methods general miss key details, namely, (a) no explicit accounting number of examples required to be covered locally, (b) no supervision for the router, which makes them myopic, and (c) evaluations only on small datasets (see Appendix §A.7 for details), etc.

**Selective Classification.** Recently, (Liu et al., 2019; Gangrade et al., 2021; Geifman & El-Yaniv, 2019) proposed learning with a reject option, wherein a model abstains from predicting uncertain instances. Although we obtain a selective classifier by ignoring the global model (see §3.2), we focus on improving the hybrid system, which is a different objective compared to this line of research.

## 2 METHOD

**Notation.** Let $\mathcal{X}$ be a feature space and $\mathcal{Y}$ a set of labels. Hybrid design consists of the following:
- A *base model* $b : \mathcal{X} \to \mathcal{Y}$, deployed on an edge device.
- A *global model* $g : \mathcal{X} \to \mathcal{Y}$ deployed on the cloud.
- A *routing model* $r : \mathcal{X} \to \{0, 1\}$ deployed on the edge.

We will treat these models as soft classifiers, outputting $|\mathcal{Y}|$-dimensional scores $\{b_y\}$ and $\{g_y\}$, and two scores $r_0$ and $r_1$ for the routing model. The hard output for the base is the top entry $b(x) = \arg\max_y b_y(x)$, and similarly for $g$. In this paper, $r$ is realized by the a 2-layer DNN with input $b_y(x)$ - this is intentionally chosen to have minimal implementation complexity. $r$ further admits a scalar tuning parameter $t$, and assigns $x$ to the global model if $r_1 > r_0 + t$, i.e.

$$r(x; t) = \mathbb{1}\{r_1(x) > t + r_0(x)\}.$$

Varying $t$ (Alg. 4) trades-off accuracy and resource usage, and allows us to avoid retraining $r$ at each resource level by locally tuning a router. By default $t = 0$. The hybrid prediction for an instance $x$ is

$$\hat{y}(x) := (1 - r(x))b(x) + r(x)g(x). \tag{1}$$

**Evaluation Metrics.** *Hybrid accuracy* is the accuracy of the hybrid predictions, defined as

$$\mathcal{A}(r, b, g) = \mathbb{P}(\hat{y}(X) = Y) = \mathbb{P}(r(X) = 0, b(X) = Y) + \mathbb{P}(r(X) = 1, g(X) = Y).$$

This accuracy is fundamentally traded-off with the *coverage* of the hybrid models, which is the fraction of instances that are processed by the cheap base model only, i.e.

$$\mathcal{C}(r, b, g) := \mathbb{P}(r(X) = 0).$$

*Modeling Resource Usage*. Coverage offers a generic way to model the resource usage of hybrid inference as follows. The resource cost of most models is mainly a function of the architecture. We let

$\alpha$ denote a generic architecture and say that $f \in \alpha$ if the function $f$ is realizable by the architecture. Then the resource cost of $f$ is denoted $\mathcal{R}(\alpha)$. Our hybrid design always executes the base and the router, and so the mean resource usage of the hybrid model $(r, b, g)$ with $b \in \alpha_b$ and $g \in \alpha_g$ is

$$\mathcal{R}(r, b, g) := \mathcal{R}_r + \mathcal{R}(\alpha_b) + (1 - \mathcal{C}(r, b, g))\mathcal{R}(\alpha_g), \qquad (2)$$

where $\mathcal{R}_r$ is a small fixed cost of executing $r$.

This generic structure can model many resources such as energy usage or total FLOPs used. Our paper is mainly concerned with *inference latency*. To model this, we take $\mathcal{R}(\alpha_b)$ to be the mean time required to execute a base model on the edge, and $\mathcal{R}(\alpha_g)$ to be the *sum of* the mean computational latency of executing $g$ on the cloud and the mean communication latency of sending examples to the cloud. In the previous example of Table 1, these numbers would be 200ms and 2025ms respectively.

**Overall Formulation.** Let $\mathscr{A}_b$ and $\mathscr{A}_g$ be sets of base and global architectures that incorporate implementation restrictions, and $\varrho$ a target resource usage level. Our objective is

$$\max_{\alpha_b \in \mathscr{A}_b, \alpha_g \in \mathscr{A}_g} \max_{r, b \in \alpha_b, g \in \alpha_g} \mathcal{A}(r, b, g) \text{ s.t. } \mathcal{R}(r, b, g) \leq \varrho. \qquad (3)$$

The outer $\max$ over $(\alpha_b, \alpha_g)$ in (3) amounts to an architecture search, while the inner $\max$ over $(r, b, g)$ with a fixed architecture corresponds to learning a hybrid model.

Below, we describe our method for solving (3). Briefly, we propose to decouple the inner and outer optimization problems in (3) for efficiency. We train hybrid models by an empirical risk minimisation (ERM) strategy. Furthermore, in Sec. 3.3, we perform architecture search using fast proxies for the accuracy attainable by a given pair of architectures without directly training hybrid models.

## 2.1 LEARNING HYBRID MODELS

This section trains hybrid models for fixed architectures $\alpha_b, \alpha_g$, i.e., the inner minimisation problem

$$\max_{r, b \in \alpha_b, g \in \alpha_g} \mathcal{A}(r, b, g) \quad \text{s.t.} \quad \mathcal{R}(r, b, g) \leq \varrho. \qquad (4)$$

Since architectures are fixed in (4), the resource constraint amounts to a constraint on the hybrid coverage $\mathcal{C}(r, b, g)$. As is standard, we will approach (4) via an ERM over a Lagrangian of relaxed losses. However, several design considerations and issues need to be addressed before such an approach is viable. These include: (a) cyclical non-convexity in (4), (b) supervision for the router, and (c) relaxed loss functions. We discuss these below and summarise the overall scheme in Algorithm 1.

**Alternating optimisation**. Problem (4) has a *cyclical non-convexity*. A given $r$ affects the optimal $b$ and $g$ (since these must adapt to the regions assigned by $r$), and vice-versa. We approach this issue by alternating optimisation. First, we train global and base models with standard methods. Then, we learn a router $r$ under a coverage penalty. The resulting $r$ feeds back into the loss functions of $b$ and $g$, and these get retrained. This cycle may be repeated.

*Modularity of training*. The scheme allows modularisation by varying which, and to what extent, steps 7,8,9 in Alg.1 are executed. It

---

**Algorithm 1** Training Hybrid Models

1: **Input:** Training data $\mathcal{D} = \{(x^i, y^i)\}_{i=1}^N$
2: **Hyper-parameters:** $\lambda_r$, Number of epochs $E$
3: **Initialize:** random $r^0$, pre-trained $b^0, g^0$.
4: **for** $e = 1$ **to** $E$ **do**
5:     Randomly Shuffle $\mathcal{D}$
6:     Generate oracle dataset $\mathcal{D}_{o;(b,g)}$ using Eq. 6
7:     $r^e = \arg\min_r \mathcal{L}_{\text{routing}}(r, b^{e-1}, g^{e-1})$
8:     $g^e = \arg\min_g \mathcal{L}_{\text{global}}(r^e, b^{e-1}, g)$
9:     $b^e = \arg\min_b \mathcal{L}_{\text{base}}(r^e, b, g^e)$
10: **Return :** $(r^E, b^E, g^E)$

---

helps train a cheap routing model to hybridise a given pair of pre-trained $b$ and $g$; or in using an off-the-shelf global model that is too expensive to train. Finally, we can learn each component to different degrees, e.g., we may take many more gradient steps on $g$ than $r$ or $b$ in any training cycle.

**Learning Routers via Proxy Supervision.** Given a base and global pair $(b, g)$, Eqn.(4) reduces to

$$\max_r \mathbb{E}[(1 - r(X))\mathbb{1}\{b(X) = Y\} + r(X)\mathbb{1}\{g(X) = Y\}] \qquad \text{s.t.} \quad \mathbb{E}[r(X)] \leq C_\varrho, \qquad (5)$$

where $C_\varrho$ is the coverage needed to ensure $\mathcal{R} \leq \varrho$. While a naïve approach is to relax $r$ and pursue ERM, we instead reformulate the problem. Observe that (5) demands that

$$r(X) = \begin{cases} 0 & \text{if } b(X) = Y, g(X) \neq Y \\ 1 & \text{if } b(X) \neq Y, g(X) = Y. \end{cases}$$

Further, while $b(X) = g(X)$ is not differentiated, the coverage constraint promotes $r(X) = 0$. Thus, the program can be viewed as a supervised learning problem of fitting the *routing oracle*, i.e.

$$o(x; b, g) = \mathbb{1}\{b(x) \neq g(x) = y\}. \tag{6}$$

Indeed, $o$ is the ideal routing without the coverage constraint. For any given $(b, g)$ and dataset $\mathcal{D} = \{(x^i, y^i)\}$, we produce the oracle dataset $\mathcal{D}_{o;(b,g)} := \{(x^i, o(x^i; b, g))\}$. It serves as supervision for the routing model $r$. This allows us to utilise the standard machine learning tools for practically learning good binary functions, thus gaining over approaches that directly relax the objective of (5).

The oracle $o$ does not respect the coverage constraint, and we would need to assign some points from the global to base to satisfy the same. From a learnability perspective, we would like the points flipped to the base to promote regularity in the dataset. However, this goal is ill-specified, and unlikely to be captured well by simple rules such as ordering points by a soft loss of $g$. We handle this issue indirectly by imposing a coverage penalty while training the routing model and leave it to the penalised optimisation to discover the appropriate regularity.

**Focusing competency and loss functions**. To improve accuracy while controlling coverage, we focus the capacity of each of the models on the regions relevant to it - so, $b$ is biased towards being more accurate on the region $r^{-1}(\{0\})$, and similarly $g$ on $r^{-1}(\{1\})$. Similarly, for the routing network $r$, it is more important to match $o(x)$ on the regions where it is 1, since these regions are not captured accurately by the base and thus need the global capacity. We realise this inductive bias by introducing model-dependent weights in each loss function to emphasise the appropriate regions.

*Routing Loss* consists of two terms, traded off by a hyperparameter $\lambda_r$. The first penalises deviation of coverage from a given target (cov), and the second promotes alignment with $o$ and is biased by the weight $W_r(x) = 1 + 2o(x)$ to preferentially fit $o^{-1}(\{1\})$. Below, $\ell$ denotes a surrogate loss (such as cross entropy), while $(\cdot)_+$ is the ReLU function, i.e., $(z)_+ = \max(z, 0)$. Sums are over training data of size $N$. Empirically, we find that $\lambda_r = 1$ yields effective results.

$$\mathcal{L}_{\text{routing}}(r; o) = \lambda_r \left( \text{cov} - \left( 1 - \frac{1}{N} \sum_x (r_1(x) - r_0(x))_+ \right) \right)_+ + \sum_x W_r(x)\ell(o(x), r(x)). \tag{7}$$

*Base Loss* and *Global Loss* are weighted variants of the standard classification loss, biased by the appropriate weights to emphasise the regions assigned to either model by the routing network - $W_b(x) = 2 - r(x)$ and $W_g(x) = 1 + r(x)$.

$$\mathcal{L}_{\text{base}}(b; r, g) = \sum W_b(x)\ell(y, b(x)); \qquad \mathcal{L}_{\text{global}}(b; r, g) = \sum W_g(x)\ell(y, g(x)).$$

## 3 EXPERIMENTS

In this section, first, we train hybrid models for resource-constrained MCU devices, thus demonstrating the effectiveness of hybrid training. Next, we show that hybrid models can be adapted to resource-rich edge devices such as mobile phones. Next, we probe various aspects of our framework through ablations, including (A) validation on other datasets, (B) sensitivity of the solution to small communication latencies, and (C) effectiveness as an abstaining classifier for situations when a global model may be unavailable. Finally, we discuss a simple joint architecture search method for finding hybrid architectures with better performance than off-the-shelf architectures (see Appendix A.3, A.4 for details).

**Experimental Setup.** We focus on the large-scale ImageNet dataset(Russakovsky et al., 2015), consisting of 1.28M train and 50K validation images. We follow standard data augmentation (mirroring, resize and crop) for training and single crop for testing. We borrow the pre-trained baselines from their public implementations (see Sec. A.6.2). Appendix A.6.1 lists our hyperparameters settings.

Throughout, we use the largest model in the OFA space (Cai et al., 2020) as our global model and henceforth refer to it as MASS-600M (which needs 600M MACs for evaluation). This model has an accuracy of 80%, and a computational latency of about 25ms on a V100 GPU. There are three main reasons for using this model. Firstly, this model is SOTA at its required computational budget. Secondly, this model is small enough for training to be tractable under our computational limitations (unlike larger models such as EfficientNet-B7, which needs 37B MACs). Finally, since it is the largest model in our architecture search space, it yields a consistent point of comparison across experiments.

To expose gains from the different hybrid components we consider: (a) 'Hybrid-(r)' - training routing models with a pre-trained base and global model, (b) 'Hybrid-(rb)' - training routing and base models

with a pre-trained global model, and (c) 'Hybrid-(rbg)' - training all three components. Unless explicitly stated, we create models by training routing and base, i.e., 'Hybrid-rb' since training the global model is computationally expensive. We train hybrid models to achieve a similar coverage as the oracle. Post-training, we tune the threshold $t$ to generate $r$ with varying coverage (Alg. 4).

*Baseline Algorithms.* We investigated prior methods (Kang et al., 2017; Nan & Saligrama, 2017; Bolukbasi et al., 2017; Li et al., 2021; Teerapittayanon et al., 2017), and the entropy thresholding, which we report here, dominates these methods across all datasets. This fact has also been observed in other works (Gangrade et al., 2021; Geifman & El-Yaniv, 2019). It is not surprising since they lack proxy supervision (2nd term in Eq. 7), and the routing model does not get instance-level supervision.

*Coverage as Proxy.* As discussed in Sec. 2, for simplicity, we use coverage as the proxy for communication latency, the major contributor to the inference latency. We explicitly model latency in Sec. 3.1, where we define the latency of the hybrid system as the sum of three latencies: (a) on-device inference, (b) communication for the examples sent to the cloud, and (c) on-cloud inference.

## 3.1 HYBRID MODELS FOR RESOURCE-LIMITED EDGE DEVICES

In this section, we train hybrid models with off-the-shelf architectures using Algorithm 1. First, we delve into the illustrative experiment in Figure 2. Next, we study a similar problem for a resource-rich edge device, specifically mobile phones, and train hybrid models at various base complexities.

**Resource Constrained MCU.** *Proposed hybrid method with MCUs for ImageNet task is near optimal (close to upper bound); realizes 25% latency/energy reduction while maintaining cloud performance, and at accuracy loss of less than 2% realizes 30% latency/energy reduction.*
Recall from Sec. 1 that we deployed an ImageNet classifier on a tiny STM32F746 MCU (320KB SRAM and 1MB Flash), the same setting as MCUNets (Lin et al., 2020). Using their TFLite model (12.79M MACs, 197ms latency, 51.1% accuracy) as the base, we create a hybrid model by adding the MASS-600 as the global model. We provide additional setup details in the Appendix §A.2.

Figure 2 shows the accuracy vs latency and energy. Table 3 shows the latency and energy of the hybrid approach against baselines. Deploying hybrid model on an MCU results in following benefits:

- *Pareto Dominance over on-device model.* Hybrid model provides $10\%$ accuracy gains over the best on-device model with similar latency. It achieves the best on-device accuracy with half the latency.
- *Pareto Dominance over other baselines.* Hybrid model achieves $5\%$ higher accuracy than the dynamic networks that use the entropy thresholding baseline (see Figure 2). In passing we recall from our earlier discussion that entropy thresholding dominates prior methods in this context.
- *Significant latency reduction to achieve SOTA accuracy.* The hybrid model achieves near SoTA accuracy with $25\%$ reduction in latency and energy consumption.
- *Micro-controller Implementation.* We deployed base and router on the MCU with negligible ($\sim 2\%$) slowdown (see Appendix A.10 for details).

Although our setup in Figure 2(a) assumes a constant communication latency of 2000ms, we can easily modify it to incorporate dynamic communication latency shown in Figure 2(b). Due to lack of space, we refer the reader to Appendix A.13 for the exact formulation.

**Resource Constrained Mobile Device.** *Gains of hybrid method are consistent, near-optimal, and dominate prior methods, across diverse devices, including large footprint devices.*
To show this, we choose the popular MBV3 (Howard et al., 2019) architectures as the base since they have been heavily deployed on mobile devices with resource constraints such as storage and compute. We create hybrid models using the following base models in the MBV3 family: MBV3-48 (2.54M params, 48M MACs), MBV3-143 (4M params, 143M MACs), and MBV3-215 (5.48M params, 215M MACs). We use MASS-600 as the global model and operate the base model at a fixed coverage level.

**Table 3:** Hybrid models on STM32F746 MCU: Accuracy achieved by different methods at various latency.

| | Accuracy (%) v/s Resource Usage | | | | | |
|---|---|---|---|---|---|---|
| Latency (ms) | 200 | 600 | 1000 | 1400 | 1600 | 2000 |
| Energy (mJ) | 11 | 53 | 96 | 140 | 161 | 216 |
| On-Cloud | - | - | - | - | - | 79.9 |
| On-Device | 51.1 | - | 60.9 | 63.5 | - | - |
| Entropy | - | 59.9 | 67.4 | 74.7 | 76.93 | - |
| Hybrid-r | - | 61.1 | 69.5 | 76.3 | 78.68 | - |
| Hybrid-rb | - | 62.0 | 70.8 | 77.7 | 79.3 | - |
| Hybrid-rbg | - | 62.3 | 71.2 | 77.9 | 79.5 | - |
| Upper-Bound | - | 62.8 | 74.7 | 79.9 | 79.9 | - |

**Table 4:** Results for hybrid models with base at various coverages. MASS-600 model achieving ≈ 80% Top1 accuracy is used as global model. Base model belongs to MBV3 space. Upper bounds (Appendix §A.2) are also reported and nearly match hybrid.

**Table 5:** Joint evolutionary search for hybrid models base constraints: 75M, 150M, & 225M. Table shows hybrid and base accuracies at different coverages. Upper bounds are reported in Appendix §A.2. Excess gains represent improved neural-network architecture.

| Base MACs | Base Acc.(%) | Method | Acc. (%) at Cov. | | |
|---|---|---|---|---|---|
| | | | 90% | 80% | 70% |
| 48M | 67.6 | Entropy | 70.7 | 73.3 | 74.9 |
| | | Hybrid | 71.6 | 74.6 | 76.8 |
| | | Upper Bnd | 72.0 | 75.2 | 78.9 |
| 143M | 73.3 | Entropy | 75.1 | 76.8 | 77.6 |
| | | Hybrid | 75.9 | 77.8 | 79.0 |
| | | Upper Bnd | 75.9 | 78.2 | 79.9 |
| 215M | 75.7 | Entropy | 77.1 | 78.3 | 78.9 |
| | | Hybrid | 77.6 | 79.0 | 79.6 |
| | | Upper Bnd | 77.6 | 79.1 | 79.9 |

| Base MACs | Base Acc.(%) | Method | Acc. (%) at Cov. | | |
|---|---|---|---|---|---|
| | | | 90% | 80% | 70% |
| 74M | 70.8 | Entropy | 73.2 | 75.3 | 76.8 |
| | | Hybrid | 74.0 | 76.4 | 78.2 |
| | | Upper Bnd | 73.9 | 77 | 79.9 |
| 149M | 74.5 | Entropy | 76.1 | 77.6 | 78.2 |
| | | Hybrid | 76.9 | 78.5 | 79.4 |
| | | Upper Bnd | 76.6 | 78.7 | 79.9 |
| 225M | 76.5 | Entropy | 77.4 | 78.3 | 79.1 |
| | | Hybrid | 78.3 | 79.6 | 80.2 |
| | | Upper Bnd | 77.8 | 79.4 | 79.9 |

Table 4 shows the hybrid accuracy at three coverage levels: 90%, 80% and 70%. Hybrid models operating at a fixed coverage provide the following benefits:

- *Up to* 70% *latency reduction to achieve SOTA.* Hybrid model with MBV3-215M base achieves 79.59% (near SOTA) with 70% coverage. It communicates with global model for only 30% inputs.
- *Hybrid models outperform the entropy thresholding.* Hybrid model with MBV3-48M achieves nearly 2% higher accuracy than entropy thresholding baseline.
- *Hybrid model with a smaller base model achieves performance of larger models at various coverage levels.* Using MBV3-48M base, the hybrid model exceeds the accuracy of MBV3-143M at 80% coverage. Similarly, it exceeds the accuracy of MBV3-215M at 70% coverage.

## 3.2 ABLATIVE EXPERIMENTS.

We refer the reader to Appendix A.11 for setup and results. Below we summarize our findings.

(A) **Other Datasets.** We train hybrid models on CIFAR-100 (Krizhevsky, 2009) and IMDb (Maas et al., 2011) (reported in Appendix), and observe similar gains.

*CIFAR-100.* We use the efficient version of the Resnet-32 (He et al., 2016) from (Dong & Yang, 2019) as the global model (35M MACs, 0.5M params, 67.31% accuracy). Base is a tiny pruned Resnet model (2.59M MACs, 0.02M params, 52.56% accuracy). Table 6 shows the performance of the hybrid models and the entropy thresholding baseline at various coverage levels. It shows that hybrid models provide similar benefits on CIFAR-100 dataset.

(B) **Other communication regimes.** We train hybrid models in a setup with no communication delay. This arises when both base and global model reside on same hardware. In such a setup, we use a simpler evaluation metric for inference latency, i.e., hybrid MACs, i.e., the amount of multiply-add operations required to execute the hybrid model. We pick up an architecture family and create a hybrid model using the smallest and largest architecture. For convenience, we use MobileNetV3 (MBV3) (Howard et al., 2019) family. From MBV3, we pick the smallest model (48M MACs, 67.6% accuracy) as the base and largest model (215M MACs, 75.7% accuracy) as global to create the Hybrid-MBV3 model. Figure 3 shows the hybrid model achieves SOTA accuracy at 30% lower latency than entropy thresholding.

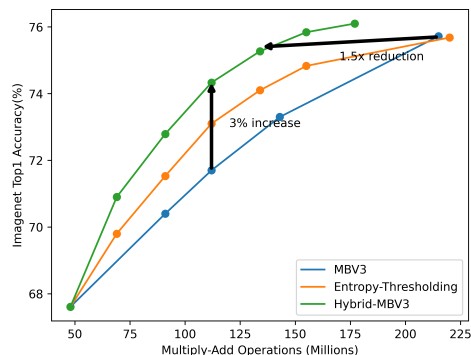

**Figure 3:** Plot for hybrid MACs vs accuracy. Base & Global models on same device.

(C) **Abstaining classifier.** When a global model is unavailable, the hybrid system can serve as an abstaining classifier on the device, i.e., it rejects a few inputs and provides predictions on the rest. We cre-

ate abstaining classifiers with the hybrid setup in the Sec. 3.1. We show the accuracy of the base model at various coverage levels in the Table 7. It shows that the abstaining base significantly outperforms the base at full coverage and the abstaining classifier from the entropy thresholding baseline, which demonstrates that our method is competitive against prior works on classification with abstention.

**Table 6:** Hybrid models for CIFAR-100 at various coverages.

| Base MACs | Base Acc.(%) | Accuracy(%) at Cov. | | |
|---|---|---|---|---|
| | | 80% | 70% | 40% |
| Entropy | 52.56 | 57.77 | 59.95 | 65.1 |
| Hybrid-r | 52.56 | 58.43 | 61.18 | 66.9 |
| Hybrid-rb | 52.56 | 59.32 | 62.48 | 67.4 |

**Table 7:** Abstaining Classifier with hybrid models from Sec. 3.1. Results for hybrid models with base at various coverage levels.

| Base MACs | Base Acc.(%) | Accuracy(%) at Cov. | | | | | |
|---|---|---|---|---|---|---|---|
| | | 90% | | 80% | | 70% | |
| | | Base | Entropy | Base | Entropy | Base | Entropy |
| 48M | 67.6 | 73.3 | 71.7 | 78.6 | 75.6 | 83.4 | 80.5 |
| 143M | 73.3 | 79.0 | 77.6 | 83.9 | 81.7 | 88.4 | 85.4 |
| 215M | 75.7 | 81.3 | 78.6 | 86.1 | 82.0 | 90.1 | 86.6 |

(D) **Router validation.** We evaluate the performance of the router against the oracle supervision and show that the router learnt using the Algorithm 1 generalizes well on the test dataset.

### 3.3 JOINT NEURAL ARCHITECTURE SEARCH FOR HYBRID MODELS.

This section proposes a joint neural architecture search (NAS) scheme to resolve the outer $\max$ problem of (3). NAS methods are strongly dependent on two aspects

**Architecture Search Space**. Our method is designed for implementation on a *Marked Architecture Search Space* (MASS). It is a set of architectures $\mathscr{A}$ such that each architecture $\alpha \in \mathscr{A}$ is associated with a known set of canonical parameters $\theta_\alpha$, which are known to be representative of the models under this architecture. That is, for most tasks, fine-tuning $\theta_\alpha$ returns a good model with this architecture. Such search spaces have been proposed by, for instance, Cai et al. (2020) and Yu et al. (2019). We use Cai et al. (2020) as search space as it contains a range of architectures of varying capacities.

**Proxy Score Function.** Given a pair of base and global architectures $(\alpha_b, \alpha_g)$, search process requires access to the value of the program (4). The score function estimates the accuracy of the hybrid model under a given resource constraint. Since training a hybrid model using these architectures would be slow, we use the proposed oracle supervision as proxy for the router. Thus, the routing oracle $o(\cdot, \alpha_b, \alpha_g)$ in conjunction with the canonical parameters $(\theta_{\alpha_b}, \theta_{\alpha_g})$ serves as an efficient proxy score function for evaluating the architecture pair $(\alpha_b, \alpha_g)$.

**Search Algorithm**. Given these above two components, NAS reduces to a combinatorial optimization and can be approached by any standard heuristic. Due to its simplicity and prevalence, we use a simple evolutionary algorithm for this (e.g Elsken et al., 2019; Liu et al., 2021) to yield a concrete method. Algorithm 2 summarizes the search scheme (see Appendix A.3 for details).

**Empirical Validation**. We evaluate the proposed architecture search scheme against the off-the-shelf architectures in Table 5 as well as Appendix A.4. It shows that evolutionary search yields higher accuracy hybrid models than off-the-shelf classifiers.

## 4 CONCLUSION

We proposed a novel hybrid design where an edge-device selectively queries the cloud only on those hard instances that the cloud can classify correctly to optimize accuracy under latency and edge-device constraints. We propose an end-to-end method to train neural architectures, base predictors, and routing models. We use novel proxy supervision for training routing models. We show that our method adapts seamlessly and near optimally across different latency regimes. Empirically, on the large-scale ImageNet classification dataset, our proposed method, deployed on an MCU, exhibits a 25% reduction in latency compared to cloud-only processing while suffering no excess classification loss.

ACKNOWLEDGMENTS

This research was supported by Army Research Office Grant W911NF2110246, the National Science Foundation grants CCF-2007350 and CCF-1955981, AFRL Contract no. FA8650-22-C-1039, and a gift from ARM Corporation.

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

# A APPENDIX

## A.1 DATASET DETAILS

**Imagenet Russakovsky et al. (2015).** This is the popular image classification dataset consisting of 1000 classes. It has nearly 1.28M training and 150K validation images. A typical image in this dataset has $224 \times 224 \times 3$ pixels. We follow standard data augmentation (mirroring, resize and crop to shape $224 \times 224$) for training and single crop for testing. Following earlier works Howard et al. (2019); Cai et al. (2020), we report the results on the validation set.

**CIFAR-100 Krizhevsky (2009).** This is a 100 way image classification dataset consisting of images with $32 \times 32 \times 3$ pixels. It has 50K train and 10K test images. We follow standard data augmentation (standard gaussian normalization/mirroring/shifting) used in earlier works He et al. (2016); Huang et al. (2017). In addition, we use the cut-out DeVries & Taylor (2017) with the standard window size of $8$.

**IMDb Maas et al. (2011).** This is a sentiment classification dataset with two classes (positive and negative). It consists of raw text of movie reviews. There are 25K train and 25K test data points. We borrow text parsers and tokenizers from the models available in the HuggingFace[2] library.

## A.2 ILLUSTRATIVE EXAMPLE DETAILS

**Edge and Cloud devices.** We use a V100 GPU as the cloud device. It has a 16GB VRAM and the server associated with this GPU has 1TB storage. We use STM32F746 [3], an ARM Cortex-M7 MCU as the edge device. It has 320KB SRAM and 1MB Flash storage.

**On-Cloud Baseline.** We use the MASS-600 Cai et al. (2020) as the global model in this experiment. It achieves $79.9\%$ Top1 accuracy on the Imagenet dataset. This model has a computational footprint of 595M MACs. As per our benchmarking, the inference latency of this model on a V100 GPU is 25ms.

**On-Device Baselines.** Lin et al. (2020) have explored deploying tiny MCUNet models on the STM32F746 MCU. We borrowed their pre-trained MCUNet models from their github repository [4] [5]. There are three different models with varying inference latencies and accuracies, namely: (a) model-A ( 12M MACs, 0.6M parameters, 200ms latency, 51.1% accuracy), (b) model-B (81M MACs, 0.74M parameters, 1075ms latency, 60.9% accuracy), and (c) model-C (170M MACs, 1.4M parameters, 1400ms latency, 63.5% accuracy). We use the smalled model, i.e., model-A as the base model in training the hybrid model. This model has the least latency as well as the least accuracy among the three models. Note that this model has the input resolution of $96 \times 96 \times 3$ and as a result the input size becomes nearly 28KB.

While we are aware of the updated version of MCUNet, i.e., MCUNetV2 (Lin et al., 2021), we point out that MCUNetV2 models are not publicly available for us to include in the empirical evaluation. Besides, our hybrid framework is completely agnostic to what the base and global model choices are. One can use any base and global model in the above setup and observe the relative gap between various baselines.

**Energy Profile Base Model Execution and Communication Cost.** We assume that the MCU device operates at 200MHz clock speed and is connected to a 3.6V power supply. It has an active mode current characteristics of $72\mu A$ per MHz. Thus, it consumes $= 3.6 \times 72 \times 200 = 51.84$ mW (i.e. mili Joules per second) energy in active mode. As a result, the energy consumed by the base model (200ms latency) on this edge device is $= 51.84 \times 0.2 = 10.368$ mili Joules. Similar calculation yield the energy consumption the on-device models: model-B (60mJ) and model-C (80mJ).

We use the NB-IoT Chen et al. (2017) communication protocol with 110kbps transmission rate and a transmission current characteristics of 30mA. Thus, it consumes $= 30 \times 3.6 = 108$ mili Joules per second in transmission mode. Let us calculate the time taken to transfer the input image from the

---

[2]https://huggingface.co

[3]https://www.st.com/en/microcontrollers-microprocessors/stm32f746ng.html

[4]https://github.com/mit-han-lab/tinyml/tree/master/mcunet

[5]https://github.com/rouyunpan/mcunet

base model on the edge device to the global model on cloud. The base model input has size 28KB and the transmission rate is 110kbps. Thus, it takes $= \frac{28*8}{110} \approx 2$ seconds to transfer this image from the edge device to the cloud. As a result, the energy consumed by the edge device in transmitting this image is $= 108 * 2 = 216$mJ.

**Split-Computation Baseline.** We use this term to refer methods Kang et al. (2017); Teerapittayanon et al. (2017) wherein the initial part of the global model executes on-device while the remaining executes on-cloud. The initial network allows for an early exit classifier for easy examples. Split into early layers results in high communication cost but split in later layers results in higher base computation latency, making it ineffective in the edge-cloud setup.

We trained the BranchyNet Teerapittayanon et al. (2017) method to represent this baseline. We create an exit classifier after the first MBConv layer in the global model and train such an exit classifier. This split results in spending 48M MACs in the base computation and then features are transferred to the global model in case the exit classifier sends the example to further layers. Note that in this case the feature cost is more than twice the input size and thus, this baseline suffers much worse from the communication delay than others. This method can always ditch the processing on the edge and send all examples to the cloud, thus achieving global accuracy with same latency and energy consumption as the on-cloud solution.

**NAS+Partition Baseline.** We use this baseline to represent methods such as LENSOdema et al. (2021) that extend split-computation by performing a neural architecture search to find the best split under the communication delay constraint. To be consistent with our evolutionary search experiments, we use the MASS to find the best split with the communication latency mentioned earlier. This search space contains architectures ranging from small to large across diverse target accuracies.

**Dynamic Neural Networks.** This baseline refers to various recent methods Nan & Saligrama (2017); Bolukbasi et al. (2017); Li et al. (2021) that propose dynamic computation and includes base-model based entropy or margin threshoding.

**Hybrid Model training.** We use the MCUNet model-A as the base model and the MASS-600 as the global model. We train the hybrid models using these base and global architecture pairs. Let $c$ be the coverage of the base model, i.e. $c$ fraction of examples are inferred on the base and $1 - c$ fraction of examples are sent to the global model. At $c = 100\%$, we obtain the on-device performance. Although hybrid models trivially achieve on-cloud performance by sending all examples to the global model, i.e. at $c = 0\%$, our hybrid training procedure obtains the on-cloud solution at with less communication. Let $\ell_b$ denote the inference latency on the edge (in this case it is 200ms), while $\ell_g$ denote the inference latency on the cloud (in this case it is the sum of communication cost and inference latency of the global model, i.e. $2000 + 25 = 2025$ms). Thus, we can write the inference latency of the hybrid model at a coverage $c$ as follows:

$$\text{Hybrid Latency@c} = \ell_b + (1 - c)\ell_g$$

Similarly, using we can compute the energy consumption for the hybrid inference as follows:

$$\text{Hybrid Energy@c} = \mu_b + (1 - c)\mu_g$$

where $\mu_b$ and $\mu_g$ are the edge device energy consumption for on-device and on-cloud inference. In this case, $\mu_b = 10.37$mJ and $\mu_g = 216$mJ.

**Upper Bound.** Let us develop an upper-bound on achievable accuracy as a function of latency. Recall from Table 1 that communication latency dominates processing time. Also recall the notion of coverage, $c$, which denotes the fraction of examples locally processed. Note that there is a one-to-one correspondence between coverage and latency of the hybrid system. Suppose $\alpha_b$, $\alpha_g$ denotes base accuracy and cloud accuracy. The base predictor predicts $(1 - \alpha_b)$ fraction incorrectly, and among these suppose we make the reasonable assumption that the router is agnostic to which of those are correctly classified at the cloud. Then an upper bound on the target hybrid accuracy is given by the expression:

$$\text{Hybrid Acc@c} \leq \min \left\{ \frac{\alpha_g - \alpha_b}{1 - \alpha_b} * (1 - c) + \alpha_b, \alpha_g \right\}$$

Notice that $c = 0$, $c = 1$, we recover global and base accuracies, and $c = \alpha_b$, is the cut-off point, namely cloud accuracy is achieved at the latency associated with the coverage $c$.

**Intuition on the Upper Bound.** The expression $c\alpha_b + (1 - c)\alpha_g$ is an underestimate of the best possible accuracy - operationally, one can see this by the fact that this is achieved simply by abstaining

randomly at the rate $c$. Quantitatively, note that $c\alpha_b + (1-c)\alpha_g \le \alpha_g$ under the natural assumption that $\alpha_g \ge \alpha_b$, and further that $c\alpha_b + (1-c)\alpha_g = \alpha_b + (1-c)(\alpha_g - \alpha_b) \le \alpha_b + \frac{1-c}{1-\alpha_b}(\alpha_g - \alpha_b)$. Let us explain our upper bound in detail:

- Let $c \ge \alpha_b$. If the router were perfect, then it would assign every point that the base model gets correct to the base - this gives us accuracy of 1 on an $\alpha_b$ fraction of inputs.
- Now we need to consider how the remaining $1 - \alpha_b$ points are assigned. Here, we make the assumption that since the router is small and cannot model the accuracy of the global model perfectly, it randomly spreads these points across the base an global models. In this case, to get overall coverage $1 - c$, it must assign a fraction of $(c - \alpha_b)/(1 - \alpha_b)$ of these remaining points to the base (in which case it we get none of these points wrong), and $(1-c)/(1-\alpha_b)$ of the remaining points to the global. We will now upper bound the accuracy of the global model on these remaining points.
- Here we use a second assumption - that every point that the base gets right is also gotten right by the global model. This is true to a good approximation, mainly due to the strong capacity difference between the models.
- This means that the conditional accuracy of the global model on the points that the base gets wrong is severely depressed - in particular, this is at most $\frac{\alpha_g - \alpha_b}{1 - \alpha_b}$ (imagine that there were $N \gg 1$ test points. The global gets $\alpha_g N$ correct. We remove the $\alpha_b N$ that the base gets right to get that there are $(\alpha_g - \alpha_b)N$ points that the global gets right and the base doesn't. Now normalise by the $(1 - \alpha_b)N$ points that the base got wrong).
- This gives us the overall accuracy

$$\alpha_b \cdot 1 + (1 - \alpha_b)\left(\frac{(c - \alpha_b)}{1 - \alpha_b} \cdot 0 + \frac{1-c}{1-\alpha_b} \cdot \frac{\alpha_g - \alpha_b}{1 - \alpha_b}\right),$$

which is exactly our upper bound (well, this, or $\alpha_g$, whichever is smaller, since the global dominates the base and so we can never get more than $\alpha_g$ accuracy).
- To sum up, the upper bound arises under the assumptions that the global model strictly dominates the base model, and that conditionally on the base model getting a query point wrong, the router is unable to meaningfully discriminate between whether the global model gets queries right or wrong.

## A.3 JOINT NEURAL ARCHITECTURE SEARCH FOR HYBRID MODELS.

**Algorithm 2** Evolutionary Joint Architecture Search

1: **Input:** Data $\mathcal{D} = \{(x^i, y^i)\}_{i=1}^N$, Architecture Search Space $\mathscr{A}$ (e.g. (Cai et al., 2020)).
2: **Hyper-parameters:** Number of generations $G$, Resource Constraint $\varphi$.
3: **Hyper-parameters:** Number of popular architecture pairs $N_{\text{pop}}$, .
4: **Hyper-parameters:** Number of popular parent architecture pairs $N_{\text{par}}$, .
5: **Initialize:** Set of popular architecture pairs $\Omega_{\text{pop}} = \{(\alpha_b^i, \alpha_g^i)\}_{i=1}^{N_{\text{pop}}}$ within constraints
6: **for** $g = 1$ **to** $G$ **do**
7:      $\Omega_{\text{par}} \leftarrow N_{\text{par}}$ highest score (Eq. 8) pairs from $\Omega_{\text{pop}}$      # Obtain the parent set $\Omega_{\text{par}}$
8:      $\Omega_{\text{child}} \leftarrow \emptyset$      # Initialize children set $\Omega_{\text{child}}$
9:      **for** $n = 1$ **to** $N_{\text{pop}}$ **do**
10:          Randomly pick $(\alpha_b^i, \alpha_g^i)$ from $\Omega_{\text{par}}$      # Pick base & global pair
11:          $(\alpha_b^m, \alpha_g^m) \leftarrow \text{Mutate}(\alpha_b^i, \alpha_g^i)$      # Add mutations on width, depth, etc.
12:          Compute the oracle $o_\varrho$ for $\theta_{\alpha_b^m}, \theta_{\alpha_g^m}$.      # Compute proxy oracle supervision
13:          **if** $\mathcal{R}(o_\varrho, \theta_{\alpha_b^m}, \theta_{\alpha_g^m}) > \varphi$ **then** GOTO 9.      # Repeat if constraint not satisfied
14:          Add $(\alpha_b^m, \alpha_g^m)$ to $\Omega_{\text{child}}$      # Add as promising child pair
15:      $\Omega_{\text{pop}} = \Omega_{\text{par}} \bigcup \Omega_{\text{child}}$      # Add children to popular set
16: **Return :** $\Omega_{\text{pop}}$      # Return promising architecture pairs

This section proposes a joint neural architecture search (NAS) scheme to resolve the outer $\max$ problem of (3). NAS methods are strongly dependent on two aspects
- *A favourable architecture space* which contains a range of architectures of varying capacities, and
- *Fast scoring proxies* that produce estimates of the overall accuracy of the best model realized by a given architecture without training the same (which would be very slow).

Given these two, NAS problems can be reduced to combinatorial optimisation and approached by any standard heuristic. In our case, we will define a favourable property of the architecture space and

then proxy scores for accuracy *at a given resource consumption*. These are plugged into a simple evolutionary search to yield a concrete method.

**Architecture Search Space**. Our method is designed for implementation on a *Marked Architecture Search Space* (MASS). It is a set of architectures $\mathscr{A}$ such that each architecture $\alpha \in \mathscr{A}$ is associated with a known set of canonical parameters $\theta_\alpha$, which are known to be representative of the models under this architecture. That is, for most tasks, fine-tuning $\theta_\alpha$ returns a good model with this architecture. Such search spaces have been proposed by, for instance, Cai et al. (2020) and Yu et al. (2019).

**Proxy Score Function.** We denote the value of program (4) for a given pair of architectures $\alpha_b, \alpha_g$ as $\mathcal{A}(\alpha_b, \alpha_g; \varrho)$. It requires a (slow) maximisation over $b \in \alpha_b, g \in \alpha_g$, as well as training a router to attain the resource constraint. Below, we describe how to construct fast viable estimates for a MASS.

*Substitute for optimisation over* $(b, g)$. MASS gives canonical models $\theta_\alpha$ for each architecture $\alpha$. Since these represent the actual performance, we take the simple approach of using the base model $\hat{b} = \theta_{\alpha_b}$ and $\hat{g} = \theta_{\alpha_g}$ to compute scores. This is the main reason we invoke the MASS concept itself.

*Substitute for optimisation over* $r$. To quickly approximate the result of training over $r$, we use the routing oracle $o$ to design a constraint-aware score.

Suppose we are given a pair of models $b, g$. Let $\mathcal{A}_o(b, g)$ and $\mathcal{C}_o(b, g)$ be the empirical accuracy and coverage of the oracle model $(o_{b,g}, b, g)$. If $\mathcal{C}_o \geq \mathcal{C}_\varrho$ then $o$ can serve as the appropriate oracle. On the other hand, if $\mathcal{C}_o < \mathcal{C}_\varrho$, we observe that we can increase the coverage of this hybrid model by simply taking a subset of $o^{-1}(\{1\})$ of relative size $\mathcal{C}_\varrho - \mathcal{C}_o$, and flipping their assignment to 0.

Importantly, which points are flipped this way is irrelevant when it comes to determining the resulting accuracy - indeed, from the perspective of $o$, switching the label of *any* point from 1 to 0 incurs the same error. Further, the accuracy that results is an upper bound on the optimal accuracy of a hybrid system satisfying the resource constraints. This gives the following proxy score for given $b, g$:

$$\mathcal{S}_o(b, g; \varrho) := \mathcal{A}_o(b, g) - (\mathcal{C}_\varrho - \mathcal{C}_o(b, g))_+ .$$

*Overall Score.* Plugging in the canonical models into the above we get the score

$$\mathcal{S}(\alpha_b, \alpha_g; \varrho) = \mathcal{S}_o(\theta_{\alpha_b}, \theta_{\alpha_g}; C_\varrho), \qquad (8)$$

which is effective under the assumptions that the architectures admit canonical models as above, and that the oracle accuracy $\mathcal{A}_o$ induces the same ordering on base-global pairs as the routing optimisation.

**Search Algorithm**. Finally, we have a scoring function and a space in hand, and so can instantiate a search algorithm. Due to its simplicity and prevalance, we propose using an evolutionary algorithm for this (e.g Elsken et al., 2019; Liu et al., 2021). Algorithm 2 summarizes the search scheme.

## A.4 EMPIRICAL VALIDATION FOR NEURAL ARCHITECTURE SEARCH OVER HYBRID SYSTEMS

*End-to-end optimization across Edge/Cloud neural architectures, predictors, and routing models achieves 80% latency reduction with an Edge model $1/4$th the size of Cloud model.*
So far, we have trained hybrid models using off-the-shelf architectures not tuned to maximize hybrid performance. Here, we search for optimised base and global pairs using the proposed evolutionary search Algorithm 2. We use the OFA space (Cai et al., 2020) as our MASS. We constrain the search to operate at fixed base MACs and coverage level 70% (on average, only 30% may be queried). Recall that coverage is a surrogate for latency in high communication latency regime. After finding base and global pairs from the evolutionary search, we create hybrid models with the newly found architectures. We perform this experiment for three edge device constraints as in the previous section: 75M, 150M, and 225M. Note that the smallest model in the MASS is close to 75M MACs. Hence, we cannot search for a model below these constraints.

Figure 4 plots the operating curves for the hybrid models found using the NAS. as well as using existing base architectures (MBV3-215M, MASS-240M). Table 5 shows the hybrid performance at the various coverage levels. Evolutionary search based hybrid models provide the following benefits:

- *Evolutionary search yields higher accuracy hybrid models than off-the-shelf classifiers.* As illustrated in Figure 4, evolutionary architecture search based hybrid models pareto dominate the hybrid models using the best known base models in MBV3 and MASS family of architectures.

- *Up to $80\%$ latency reduction to achieve SoTA accuracy.* Using a base with 225M MACs and $76.5\%$ accuracy, the hybrid model achieves $79.6\%$ accuracy at $80\%$ coverage.
- *Hybrid models outperform entropy thresholding.*

## A.5 Algorithms

We summarise the methodological proposals as algorithms. The overall method is to begin with training a super-net in the sense of Cai et al. (2020), for which the methods of their paper can be utilised. This produces a set of architectures $\mathscr{A}$, with associated canonical models for each $\alpha \in \mathcal{A}$. The overall procedure then is summarised as Algorithm 3. This uses the two main procedures of architecture search (Algorithm 2) and hybrid training (Algorithm 1) as subroutines, which in turn may be executed in a modular way as discussed at length in the main text.

In addition, we frequently tune a given router $r$ and base and global models to locally trade-off resource usage levels and accuracy (which saves on retraining on each different value of $\varrho$ that one may be interested in. This is realised by finding a value $t$ adjusted to the constraint, and using the routing function $r(x;t) = \mathbb{1}\{r_o(x) \geq r_1(x) + t\}$. Such a $t$ may be found as in Algorithm 4.

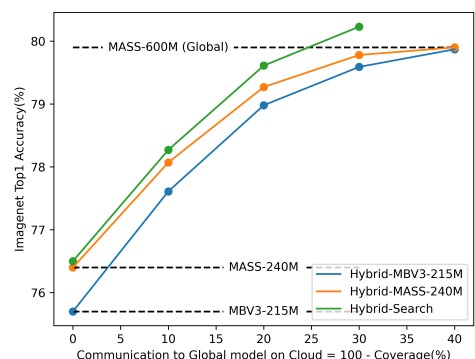

**Figure 4:** Comparing hybrid models trained with Off-the-Shelf architectures vs architectures found using Joint NAS (Algorithm 2).

---

**Algorithm 3** End-to-end Hybrid Procedure

---

1: **Input:** Training data $B = \{(x^i, y^i)\}_{i=1}^N$, Validation data $V = \{(x^j, y^j)\}_{j=1}^M$, resource constraint $\varrho$.
2: Train supernet using the method of Cai et al. (2020).     *(Architecture Search)*
3: $\mathscr{A} \leftarrow$ resulting set of algorithms.
4: $(\alpha_b, \alpha_g) \leftarrow$ output of Algorithm 2 with $V, \varrho, \mathscr{A}$.
5: Train initial models $b^0 \in \alpha_b, g^0 \in \alpha_g$ using $B$     *(Hybrid Training)*
6: $(r, b, g) \leftarrow$ output of Algorithm 1 instantiated with $B, b^0, g^0$, and with appropriate hyperparameters.
7: **Return:** $(r, b, g)$

---

**Algorithm 4** Tuning Routing Model

---

1: **Input:** Validation data $V = \{(x^j, y^j)\}_{j=1}^M$, target resource level $\varrho$, Hybrid model $(r, b, g)$.
2: $\mathcal{T} \leftarrow \{r_0(x) - r_1(x) : x \in V\}$.
3: $c^* \leftarrow \min c : \mathcal{R}_r + \mathcal{R}(\alpha_b) + (1 - c)\mathcal{R}_g \leq \varrho$.
4: $t^* \leftarrow c^*$th quantile of $\mathcal{T}$.
5: **Return:** $t^*$.

---

## A.6 Implementation Details

### A.6.1 Hyper-parameter Settings.

We use SGD with momentum as the default optimizer in all our experiments. We initialize our hybrid models from the corresponding pre-trained models and use a learning rate of $1e-4$ for learning base and global models. We use a learning rate of $1e-2$ for learning the routing network. We decay the learning rate using a cosine learning rate scheduler. As recommended in the earlier works, we use a weight decay of $1e-5$. We set the number of epochs to be 50. We use a batch size of 256 in our experiments.

### A.6.2 MODEL DETAILS

**Entropy Thresholding Baseline.** As per recommendation in the literature (Teerapittayanon et al., 2017; Gangrade et al., 2021) we compute the entropy $H$ of the base prediction probability distribution $b_y(x)$. This baseline allows access to a tunable threshold $t$. Predictions with entropy below this threshold are kept with the base model while the predictions with entropy above this threshold are sent to the cloud model. We use similar tuning as Algorithm 4 to trade-off resource usage.

**Routing Model.** Our routing model uses predictions from the base model and creates a 2-layer neural network from these predictions. We create meta features from these predictions to reduce the complexity of the network, by (a) adding entropy as a feature, (b) and adding correlations between top 10 predictions, resulting in a 101 dimensional input feature vector. The feed-forward network has 256 neurons in the first and 64 neurons in the second layer. The final layer outputs a two dimensional score leading to binary prediction for the routing $r$. Note that the routing network described in this manner contributes to less than 2% compute budget of the base model and hence its compute cost is negligible in comparison to the base and global models.

**MBV3.** We have used the MobileNetV3 Howard et al. (2019) models as base in the hybrid models designed for mobile devices (see Sec. 3.1 ). We borrowed pre-trained models from publicly available implementation [6]. Table 8 lists the performance and compute characteristics of these borrowed models.

Table 8: MBV3 models in our setup.

|          | Top1 Accuracy | #Params | #MACs  |
|----------|---------------|---------|--------|
| MBV3-48  | 67.613        | 2.54M   | 48.3M  |
| MBV3-143 | 73.3          | 3.99M   | 143.4M |
| MBV3-112 | 71.7          | -       | 112M   |
| MBV3-91  | 70.4          | -       | 91M    |
| MBV3-215 | 75.721        | 5.48M   | 215.3M |

**MASS.** We borrowed the pre-trained Cai et al. (2020) models from the official public repository [7]. Table 9 lists the accuracy, number of parameters and MACs for these models. We note that these models have been specialized by the authors with fine-tuning to achieve the reported performance.

Table 9: Once-for-All Pre-trained models in our setup.

|                                                      | Top1 Accuracy | #Params | #MACs |
|------------------------------------------------------|---------------|---------|-------|
| MASS-600 ('flops@595M_top1@80.0_finetune@75')        | 79.9          | 9.1M    | 595M  |
| MASS-482 ('flops@482M_top1@79.6_finetune@75')        | 79.6          | 9.1M    | 482M  |
| MASS-389 ('flops@389M_top1@79.1_finetune@75')        | 79.1          | 8.4M    | 389M  |
| MASS-240 ('LG-G8_lat@24ms_top1@76.4_finetune@25')    | 76.4          | 5.8M    | 230M  |
| MASS-151 ('LG-G8_lat@16ms_top1@74.7_finetune@25')    | 74.6          | 5.8M    | 151M  |
| MASS-101 ('note8_lat@31ms_top1@72.8_finetune@25')    | 72.8          | 4.6M    | 101M  |
| MASS-67 ('note8_lat@22ms_top1@70.4_finetune@25')     | 70.4          | 4.3M    | 67M   |

### A.7 DIFFERENCE BETWEEN APPEALNET AND OUR HYBRID DESIGN.

Below we highlight main difference between AppealNet (Li et al. (2021)) and our proposal.

- AppealNet formulation does not explicitly model any coverage constraint that enables the base model to operate at a tunable coverage level. In contrast, we explicitly model a coverage penalty.

- Jointly learning the routing without any supervision is a hard problem. Instead, we relax this formulation by introducing the routing oracle that specializes in a routing network for a given base and global pair. With this oracle, the task of learning routing reduces to a binary classification

---

[6]https://github.com/rwightman/pytorch-image-models
[7]https://github.com/mit-han-lab/once-for-all

problem with the routing labels obtained from the oracle. This also decouples the routing task from the base and global entanglement.

- AppealNet does not use supervision like us, and as such such strategies ultimately resemble thresholding on examples whose anticipated loss exceeds some threshold. To see this consider Algo. 1 line 7 Li et al. (2021) for $m$ examples. Taking the derivative wrt $q$ yields $\beta \frac{1}{m} \sum_x \frac{1}{1-q(0|x)} = \frac{1}{m} \sum_x \ell(f_1(x), y) - \ell(f_0(x), y)$. The RHS is the *excess loss*. For small values of $q(0|x)$, we can approximate the LHS to yield: $\beta \frac{1}{m} \sum_x (1 + q(0|x)) \approx$ Excess-Loss. Simplifying we get: $\frac{1}{m} \sum_x q(0|x) \approx \frac{1}{\beta}$Excess-Loss $- 1$. This expression suggests a relaxed objective that we should enforce the fact that examples sent to the cloud is broadly proportional to excess loss, and as such represents very weak supervision.

- In addition, we propose a neural architecture search that finds a pair of base and global architectures that optimise the hybrid accuracy at any given combined resource usage.

- Empirically, AppealNet does not have any evaluations for the Imagenet scale dataset. The closest comparison we can find is with the Tiny-Imagenet dataset (one-tenth of the size of the Imagenet). While we cannot compare the two directly, since we solve a much harder problem than Tiny-Imagenet, we can make the following observations. At $70\%$ coverage level, for AppealNet, the minimum performance difference between the hybrid model and the global model is $\approx 1.2\%$ (see AppealNet, Fig. 5(d)), while our closest to the global in case of the MobileNet baseline is $0.3\%$ (see our paper Table 1, row 3). Note that AppealNet performance will go down on Imagenet in comparison to Tiny-Imagenet due to the hardness of the problem.

- We compare Entropy Thresholding, AppealNet, and the proposed Hybrid model on the CIFAR-100 dataset. Table 10 shows that entropy thresholding outperforms AppealNet. In addition, the proposed Hybrid model outperforms these baselines substantially.

**Table 10:** Hybrid models for CIFAR-100 at various coverages.

| Scheme | Base Acc.(%) | Accuracy(%) at Cov. | | |
|---|---|---|---|---|
| | | 80% | 70% | 40% |
| Entropy | 52.56 | 57.77 | 59.95 | 65.1 |
| AppealNet | 52.56 | 57.65 | 59.89 | 64.9 |
| Hybrid-r | 52.56 | 58.43 | 61.18 | 66.9 |
| Hybrid-rb | 52.56 | 59.32 | 62.48 | 67.4 |

## A.8 DIFFERENCE BETWEEN LENS AND OUR HYBRID DESIGN.

Although below we highlight main difference between LENS (Odema et al. (2021)) and our proposal, we emphasize that LENS studies the edge-cloud interactions purely from systems perspective and as such does not dwelve into the learning aspects and the trade-off required in routing the inputs in severely resource constrained edge devices as well as their limited communication capabilities.

- (A) **Our Objective:** On large-scale tasks (such as ImageNet), for the given WiFi rate, our goal is to realize the accuracy of an arbitrarily large DNN model (deployable on the cloud) by means of a hybrid method deployed on edge. We selectively route difficult inputs on the edge to the cloud, maintaining top-accuracy, while consuming minimal energy/latency.

  *Our Edge.* Our edge device only has CPU or MCU compute capabilities (see Sec. 3.1). In addition, these edge device are severely resouce constrained, namely they only allow low powered transmission as well as low transmission rates. These constraints limits the model that can be deployed on the edge to be very low footprint. For instance, our illustrative example only has 110Kbps transmission rate (see Sec. A.2).

  This together with our desired accuracy places stringent constraints on edge to crisply learn hard-to-predict examples, and characterizes **fundamental limits of hybrid ML**.

- (B) **Circuit/Systems Prior Works (ex: LENS or Neurosurgeon).** Their goal for a given dataset is to split/partition computation of a (suitably optimized) DNN to minimize latency/energy in response to changing wireless conditions.

*LENS Edge.* Odema et al. (2021) has GPU compute capabilities on the edge device. As a result, even large DNNs can be comfortably executed on this device without a significant delay as compared to the on-cloud solution. In addition, their edge device leverage high transmission rate (up to 25Mbps). As a result, LENS explores a different setting **agnostic** to data. They leverage low transmission latency to compensate the difference in edge and cloud GPU times, motivating partitioning.

- (C) **Objective (B) is suboptimal for objective (A).** (B) requires the same network model on the edge and the cloud, which artificially constrains DNNs to fit into edge's specifications, while hoping to realize cloud-server gains on the partitioned parts. For large-scale tasks (ImageNet), high-accuracy can only be achieved by large DNNs (even with NAS optimization Cai et al. (2020)), which are not edge-deployable, and using different architectures on edge/cloud is fundamental in (A).

- (D) **LENS baselines are too weak.** Direct comparisons are difficult due to different system choices(see (B)). Still we can note that LENS:

    - reports results on small CIFAR-10 dataset, which is not representative
    - uses VGG-16 based architecture(large DNN)—typically not edge-deployable
    - with all-cloud processing, achieves 82%, significantly lower than VGG16 published results (93.56% see `https://github.com/geifmany/cifar-vgg`);
    - with optimal NAS+partitioning gets 77% under 25% energy reduction (see Fig. 6)

    In contrast, for CIFAR-10 we trained standard (tiny) models (see MCUNet model described as On-Device baseline in Sec. A.2) deployable on MCUs.

    - MCUNet Lin et al. (2020) is deployable on the resource constrained MCUs.
    - VGG-16 on CPU has 280X worse latency w.r.t. our model
    - With all-edge processing we get 91.7% accuracy consuming 11mJ, and 85% under 25% energy reduction.

    This shows using same model on cloud/edge is suboptimal (see (C)).

- (E) **Large-scale Task: LENS code is unavailable; Direct Evaluation is difficult.** LENS employs GP-based bayesian optimization to NAS, which is known to produce poor results (see White et al. (2021)), which is also evident from (D). Due to lack of publicly avaiable codebase, we created a similar baseline to see the performance gap between our proposal and LENS on the illustrative example in the introduction (see Figure 2). We optimized NAS+partitioning method by optimizing over OFA models/architectures Cai et al. (2020). These architectures range from small to large models across diverse target accuracies. Hybrid methods overwhelmingly dominate NAS+partitioning methods (pink in Figure 5), again reinforcing our point (C).

### A.9 ONCE-FOR-ALL SEARCH EXPERIMENTS

For our evolutionary search experiments, we used the Cai et al. (2020) as the MASS. In this space, there are two MobileNetV3 backbones available: (a) one with width multiplier 1 and (b) another with width multipler 1.2. The range of models in these two space together is around $75 - 600M$ MACs. MASS allows searching over expansion factor options [3,4,6], width multiplier [1, 1.2], convolutionarl kernel sizes [3,5,7], block depths [2,3,4], and resolutions [144, 160, 176, 192, 208, 224].

To perform a mutation, each optimization variable is modified with probability $0.1$, where modification entails re-sampling the variable from a uniform distribution over all of the options. The population size is set to 100, and the parent set size is set to 25.

Table 11 shows the characteristics of the base and global models found using this search. Similar to Cai et al. (2020) we fine tune these models further for $50$ epochs with their setup to achieve the final accuracy.

### A.10 MCUNET ROUTER DEPLOYMENT OVERHEAD

We deploy both MCUNet and our base with routing model on the MCU using the TensorFlow Lite for Microcontrollers (TFLM) runtime. Due to lack of operator support for reductions and sorting in TFLM, we replace the relevant operators with supported operations whose compute and memory

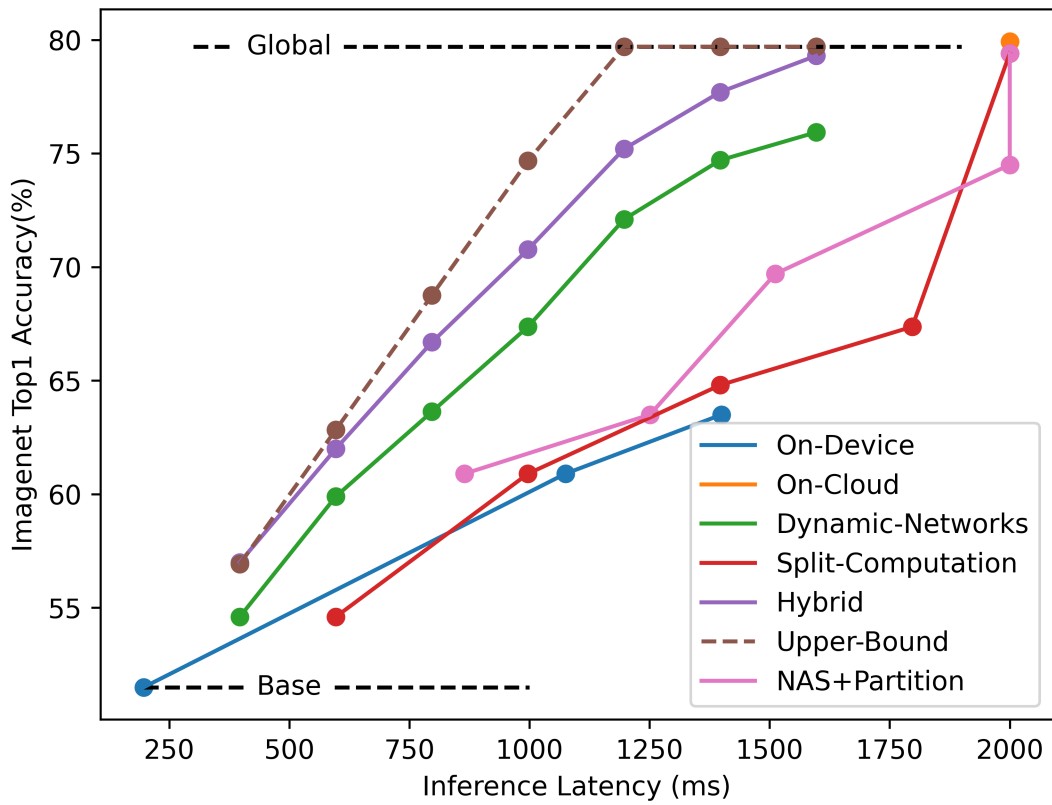

**Figure 5:** Image recognition on the ImagetNet dataset: Accuracy vs Energy and Latency plot. This clearly shows that the hybrid design pareto dominates on-device as well as other baselines while getting significantly closer to the upper-bound in hybrid design.

**Table 11:** Joint Evolutionary Architecture Search: Models found at three different base MAC constraints (75M, 150M, 225M).

|  | Top1 Accuracy | #Params | #MACs |
|---|---|---|---|
| Search-MASS-225 | 76.5 | 5.8M | 225M |
| Search-MASS-149 | 74.5 | 5.3M | 149M |
| Search-MASS-75 | 70.8 | 4.5M | 75M |

complexity upperbounds the un-supported operations. Table 12 compares the performance energy profile of the hybrid model and the baseline when deployed on the micro-controller (STM32F746) with 320KB SRAM & 1MB Flash. It clearly shows that there is a negligible cost of deploying the proposed routing scheme and only results in $< 2\%$ slowdown.

**Table 12:** Profiling the on device latency and energy overhead associated with deploying the Hybrid model (MCUNet + router) as compared to deploying the plain MCUNet model on the MCU.

| Model | Latency | SRAM | Energy |
|---|---|---|---|
| MCUNet | 0.25368s | 156708 bytes | 0.1112 joules |
| Hybrid-MCUNet | 0.25951s | 158036 bytes | 0.1134 joules |

## A.11 ABLATIVE EXPERIMENTS

### A.11.1 BASE AND GLOBAL ON SAME DEVICE

So far we have focused on a setup where base and global models are deployed on separate hardware. In this experiment, we deploy the base and global models on the same device. As a result, there is no communication delay in the setup. In such a setup, we can use a simpler evaluation metric for inference latency, i.e., hybrid MACs, i.e., the amount of multiply-add operations required to execute the hybrid model. We pick up an architecture family and create a hybrid model using the smallest and largest architecture. For convenience, we perform this experiment for a known family, namely MobileNetV3 (MBV3) (Howard et al., 2019). From MBV3, we pick the smallest model (48M MACs, 67.6% accuracy) as the base and largest model (215M MACs, 75.7% accuracy) as global to create the Hybrid-MBV3 model. Figure 6 shows the hybrid model performance against the intermediate points in the MBV3 space.

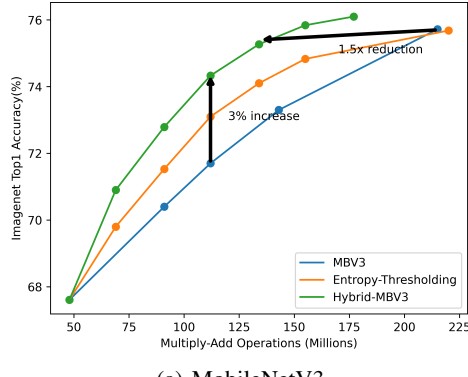

(a) MobileNetV3

**Figure 6:** MBV3: Plot for hybrid MACs vs accuracy.

Figure 6 shows the hybrid model performance against the intermediate points in the MBV3 space as well as entropy thresholding baseline. These experiments provide evidence for the following properties of hybrid models:

- *Hybrid achieves SOTA w.r.t a global model with up to* 40% *lower latency.* Global model in MBV3 achieves 75.7% accuracy by off-loading every example to the cloud while hybrid model achieves same accuracy by sending only 60% examples to the cloud. Thus, saving 40% communication cost.

- *Training a Hybrid model for intermediate latency is inexpensive.* To achieve a single model at any latency, we find an architecture with this constraint and train it to non-trivial performance. Hybrid model with extreme points trades off latency for accuracy and save compute for training models for any intermediate constraint.
- *Hybrid models dominate entropy thresholding baseline used in dynamic neural networks.* Hybrid models outperform entropy thresholding at every coverage level with up to $1.5\%$ accuracy gains.

### A.11.2  Router validation

We evaluate the performance of the router against the oracle supervision and show that the router learnt using the procedure described in Sec. 2.1 generalizes well. For instance, while training a hybrid model with pre-trained MBV3-small and MBV3-large models, on the oracle labels, the router achieves a training accuracy of $\approx 87\%$ and this translates into a validation accuracy of $\approx 84\%$. In contrast, entropy thresholding on the validation dataset achieves $\approx 77\%$ accuracy on the oracle labels.

### A.11.3  IMDb Experiments

We also learn hybrid model in the NLP domain. We train a sentiment classifier on the IMDb dataset Maas et al. (2011). We pick-up off-the-shelf pre-trained classifiers, namely (a) albert-base-v1 Lan et al. (2019) (11M params, $91\%$ accuracy) as the base and (b) bert-large Devlin et al. (2018) (340M params, $96\%$ accuracy) as the global. We use the hidden state from the last timestep in the sequence along with the classifier logits and entropy as the feature for the routing model. In order to save computation, we learn the hybrid model by training only the router. Table 13 shows the performance of the hybrid models and the entropy thresholding baseline at various coverage levels. It shows that hybrid models provide similar benefits on IMDb dataset. We note that, for further model footprint reduction, similar hybrid models can be constructed using efficient recurrent neural networks Kag et al. (2020); Kag & Saligrama (2021) as the base model and large transformer models as the global model.

**Table 13:** Hybrid models for IMDb at various coverages.

| | Base MACs | Base Acc.(%) | Cov.=97% Acc. (%) | Cov.=95% Acc. (%) | Cov.=93% Acc. (%) |
|---|---|---|---|---|---|
| Entropy | 91 | 93.78 | 94.38 | 95.25 | |
| Hybrid | 91 | 94.31 | 95.45 | 96.01 | |

### A.11.4  MCUNet experiment with EfficientNet-B7

In the main text, due to limited compute resources, we restricted our global model to be the MASS-600 Cai et al. (2020) model. In this ablation, we explore the effect of deploying a significantly expensive model on the cloud. We choose the best performing model in the EfficientNet Tan & Le (2019) family, i.e., EfficientNet-B7. This model has 37B MACs and stores 66M network parameters. We borrow the implementation from the timm repository [8] that achieves an accuracy of $86.5\%$ on the Imagenet classification task. In contrast, the MASS-600 model has $\approx 600$M MACs, 9.1M parameters and achieves $\approx 80\%$ accuracy. For simplicity, we assume the cloud resources render the inference on EfficientNet-B7 to be similar to MASS-600. In this experiment, we train a hybrid-r model with MCUNet base used in the Sec. 3.1. Thus, we can compare the performance of the hybrid models across different global models. Table 14 compares the accuracies obtained by the hybrid models with two different global models (MASS-600 and EfficientNet-B7). It clearly shows the following benefits:

- Deploying a better global model improves the hybrid performance and with cloud resources such large models do not affect the energy consumption on the edge device.
- Assuming that the cloud has access to large compute pool, the inference latency on the edge device does not suffer as well.

---

[8] EfficientNet-B7-ns model from https://github.com/rwightman/pytorch-image-models

- It shows that our algorithm procedure to train hybrid models works across global models in different architecture families.

**Table 14:** EfficientNet-B7 as Global model: Hybrid models on STM32F746 MCU: Accuracy achieved by different methods at various latency constraints. Base model is the MCUNet model with 12M MACs and 200ms latency.

| Method | Accuracy (%) at Latency (ms) | | | | | |
|---|---|---|---|---|---|---|
| | 200 | 600 | 1000 | 1400 | 1600 | 2000 |
| On-Cloud (MASS-600) | - | - | - | - | - | 79.9 |
| On-Cloud (EfficientNet-B7) | - | - | - | - | - | 86.5 |
| On-Device | 51.1 | - | 60.9 | 63.5 | - | - |
| Hybrid (Global=MASS-600) | - | 62.3 | 71.2 | 77.9 | 79.5 | - |
| Hybrid (Global=EfficientNet-B7) | - | 64.9 | 76.2 | 82.8 | 85.7 | - |

## A.12 SOCIETAL IMPACT AND LIMITATIONS

Edge devices offer a range of predictive services to end users. Often times, these services require access to cloud resources to guarantee state-of-the-art performance as the edge-deployable model is very restrictive. Such a cloud interaction has many downsides, including (a) increased inference latency, (b) increased battery usage on edge, (c) additional cost for cloud resources, (d) end user privacy concerns, and (e) increased carbon footprint due to cloud usage.

The proposed hybrid design aims to address these issues in the existing on-cloud solutions by only selectively querying the cloud on inputs that are hard to classify locally. Our proposal engages the cloud only when it is sufficiently confident that the cloud usage would result in correct classification. As a result, it increases the coverage, i.e., the number of inputs locally predicted by the base model without invoking the cloud model. Thus, the overall cloud utilization is reduced significantly as seen from our empirical evaluations. Note that, in the worst case, when the input is hard to predict locally, the router engages the cloud and gets hit with on-cloud latency.

Our hybrid design, as much as possible, helps the existing predictive services to selectively querying the cloud. As a result, we get the following benefits as compared to the existing on-cloud solutions:

1. Decreased inference latency on the edge device due to communication delay
2. Decreased battery/energy usage on the edge device due to data transmission
3. Reduced cost for access to cloud resources
4. Reduced carbon footprint due to less resource usage
5. End user privacy concerns are addressed in two ways. First, by selectively querying the cloud, only a small fraction of data is sent to the cloud. Second, hybrid design allows the base model to operate as the abstaining classifier and in the process enables the practitioners to ask for user consent by showing the base model prediction and router confidence in those predictions. Such a design gives the users informed consent on whether they are satisfied with the inference results or they would like to send the data to cloud as their local models are unable to correctly classify the inputs.

## A.13 DYNAMIC COMMUNICATION LATENCY

Although our setup in Figure 2 assumes a constant communication latency, we can easily modify the setup to incorporate dynamic latency delay. Assuming the base processing latency $B$, global processing latency $G$, communication delay $D$, and router coverage $C$, we can use the following constraint on the coverage $C$ to achieve a target average latency $L$,

$$B + (1 - C) * (D + G) \leq L$$

As per the main text, $B = 200$ms, $G = 25$ms and the communication delay $D = 2000$ms. In case the communication delay $D$ has a high variation and ranges between $[D_{\min}, D_{\max}]$, we can store a lookup table on the edge device to use a different coverage threshold for different observed latency, by solving the above constraint at pre-defined communication delay intervals.

For the illustrative example, we simulate this setting by drawing communication latencies uniformly at random between $[D_{\min}, D_{\max}]$. Figure 7 compares the dynamic communication with the constant communication cost setting. It has three dynamic ranges [200, 2000], [1000, 3000], [500, 3500],

where the last two range keep the mean communication cost same as the constant setting. It can be clearly seen that proposed hybrid models outperform all the baselines in each setting.

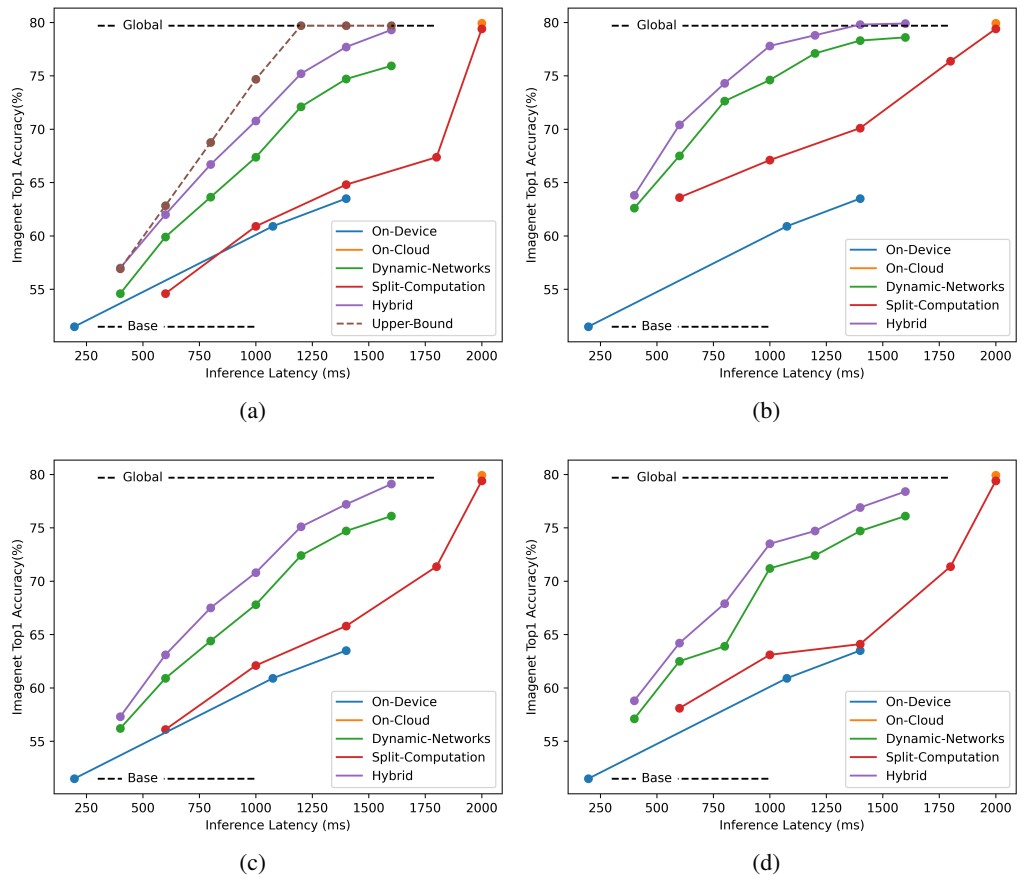

**Figure 7:** Setup is same as Figure 2: (a) Constant Communication Latency (b) Dynamic Communication latency $[400, 2400]$ (c) Dynamic Communication latency $[1000, 3000]$ (d) Dynamic Communication latency $[500, 3500]$

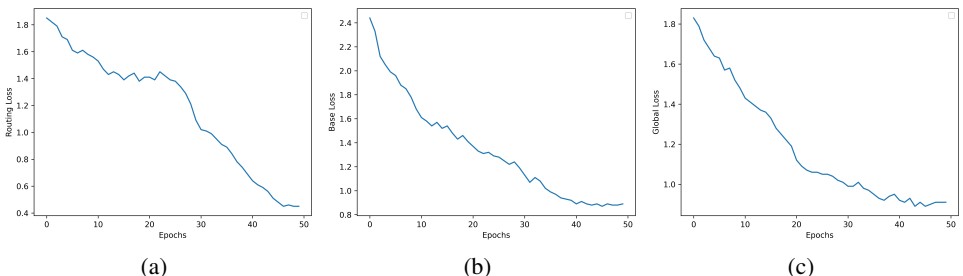

**Figure 8: Algorithm Convergence.** We show the training losses for the three components in the Algorithm 1: (a) router, (b) base, and (c) global model.

## A.14   ALGORITHM CONVERGENCE ANALYSIS

In this ablation, we show that the Algorithm 1 for training hybrid models convergences empirically. We pick up the base and global model as described in the setup in the Figure 3. We plot the training losses for the three components (router, base, global models) in the Figure 8. It shows that base and global models convergence to a stable loss towards the end of the training cycle.

