# OpenReview forum: "Efficient Edge Inference by Selective Query"
_ICLR.cc/2023/Conference — ICLR 2023 poster_

### Official Review · Reviewer_XmVX · 2022-10-24

**Confidence:** 4
**Clarity, Quality, Novelty And Reproducibility:** 1. The main paper is incomplete. The …
**Correctness:** 3
**Technical Novelty And Significance:** 2
**Empirical Novelty And Significance:** 3
**Recommendation:** 6

**Strength And Weaknesses:**

Strength:
1. The proposed criteria for label construction for router training is reasonable and efficient. The criteria exclude corner cases in which cloud and device models fail.
2. The joint training of models on cloud and local devices is impactful. Besides training a router to deliver a sample, the authors find that training each model (on a cloud or local device) can also benefit from such a router. This resolution leads a jointly trained base model to outperform an individually trained one experimentally, even with some abstaining.
3. The paper is well-organized. The authors highlight the proposed scheme and provide several model structure optimization solutions.
4. The experiments in this work are sufficient.

Weakness:
1. The metric used for the "routing model" is unavailable for regression tasks. The metric includes a strict equivalence evaluation between labels and predictions to identify the target execution platforms. However, the equivalence is unavailable in regression.
2. The upper bound accuracy calculation is unreasonable. First, the proportion $\frac{1 − c}{1 − a_b}$ in the formulation of upper bound accuracy is weird. Due to loss function in Eq (7), if optimality holds, $P(r(x) = 0)$ should be equal to $c$. Due to Eqn (1) and related statements, the upper bound should be $c a_b + (1-c)a_g$ since execution on device and cloud are complementary. Furthermore, the authors assume that accuracies on cloud and local devices are unchanged. However, this is unsustainable since distributions change in the proposed hybrid training framework. For example, the loss functions for device and cloud under Eqn (7) erase some samples for both.
3. Some conclusions in the method section lack theoretical or experimental demonstrations. For example, there is no convergence demonstration of proposed algorithm 1. In the loss functions for cloud and device, several experiments to demonstrate the effectiveness of these "hard masks" will make it more reasonable.



**Summary Of The Paper:**

This paper proposed a supervised training framework to improve the classification model inference in edge-cloud cooperating scenarios. The framework utilizes pre-trained models to achieve labelization for the router model under the criteria that the samples that should be transferred to the cloud are cases in which local models fail. Moreover, the authors also utilize the trained router model to re-balancing the importance of samples for classification models on cloud and device. Optimizing the router model and classification are conducted jointly to improve each system component. Extensive experimental results on benchmarks demonstrate the outperformance of SOTA.



**Summary Of The Review:**

This paper applies a data-based mixture of expert method on edge-cloud computing scenarios. The main contribution is that the mixture is conducted by data, simultaneously solving the model selection and improvement. Although some details require a more delicate design and the current approach is limited to the classification scenario, the proposed alternative training framework may facilitate delicate and in-depth investigations in the future.

---

> ### Author Response · Authors · 2022-11-16
> **Convergence demonstration of the proposed Algorithm 1**
>
> In the revision, we have added convergence plots for all the components in the Algorithm-1 in the Appendix A-14.  Figure-8 shows that router, base and global models converge to a small loss value.

---

> ### Author Response · Authors · 2022-11-16
> **Upper bound on hybrid accuracy is unreasonable**
>
> The expression $c \alpha_b + (1-c)\alpha_g$ is an underestimate of the best possible accuracy - operationally, one can see this by the fact that this is achieved simply by abstaining randomly at the rate $c$. Quantitatively, note that $c \alpha_b + (1-c) \alpha_g \le \alpha_g$ under the natural assumption that $\alpha_g \ge \alpha_b$, and further that $c \alpha_b + (1-c)\alpha_g = \alpha_b + (1-c)(\alpha_g - \alpha_b) \le \alpha_b + \frac{1-c}{1-\alpha_b}(\alpha_g - \alpha_b).$
>
> Let us explain our upper bound in detail:
> - Let $c \ge \alpha_b.$ If the router were perfect, then it would assign every point that the base model gets correct to the base - this gives us accuracy of $1$ on an $\alpha_b$ fraction of inputs.
> - Now we need to consider how the remaining $1-\alpha_b$ points are assigned. Here, we make the assumption that since the router is small and cannot model the accuracy of the global model perfectly, it randomly spreads these points across the base an global models. In this case, to get overall coverage $1-c$, it must assign a fraction of $(c - \alpha_b)/(1-\alpha_b)$ of these remaining points to the base (in which case it we get none of these points wrong), and $(1-c)/(1-\alpha_b)$ of the remaining points to the global. We will now upper bound the accuracy of the global model on these remaining points.
> -  Here we use a second assumption - that every point that the base gets right is also gotten right by the global model. This is true to a good approximation, mainly due to the strong capacity difference between the models.
> - This means that the conditional accuracy of the global model on the points that the base gets wrong is severly depressed - in particular, this is at most $\frac{\alpha_g - \alpha_b}{1-\alpha_b}$ (imagine that there were $N \gg 1$ test points. The global gets $\alpha_g N$ correct. We remove the $\alpha_b N$ that the base gets right to get that there are $(\alpha_g - \alpha_b)N$ points that the global gets right and the base doesn't. Now normalise by the $(1-\alpha_b)N$ points that the base got wrong).
> - This gives us the overall accuracy $$ \alpha_b \cdot 1 + (1-\alpha_b) \left(\frac{(c-\alpha_b)}{1-\alpha_b}\cdot 0 + \frac{1-c }{1-\alpha_b} \cdot \frac{\alpha_g - \alpha_b}{1-\alpha_b}\right),$$ which is exactly our upper bound (well, this, or $\alpha_g,$ whichever is smaller, since the global dominates the base and so we can never get more than $\alpha_g$ accuracy).
> - To sum up, the upper bound arises under the assumptions that the global model strictly dominates the base model, and that conditionally on the base model getting a query point wrong, the router is unable to meaningfully discriminate between whether the global model gets queries right or wrong.

---

> ### Author Response · Authors · 2022-11-16
> **Routing model unavailable for regression tasks.**
>
> Regression is not the focus of this work. We focus only on the classification task as it is an important problem on its own.

---

> ### Author Response · Authors · 2022-11-16
> **Clarity: 2.1 : In the introduction section, the authors fail to demonstrate the potential advantages of joint optimizations on the server, edge devices, and router.**
>
> - In the introduction, Figure 2 shows that hybrid models outperform the Dynamic Neural Networks. This clearly shows the advantages of the joint optimization on the server, edge devices and router.
>
> - A post-hoc solution does not work (as illustrated in Figure 2, and Table 3,4,5) simply because it learns a routing strategy agnostic to global model predictions and correctness. In our proposal, the proxy oracle routes the examples to the global model only when the global model is correct and the base is incorrect.

---

> > ### Author Response · Authors · 2022-11-16
> > **Clarity 2.2: In the methods section, the authors present a simple supervised solution for the router model in a complicated manner. Most mathematical formulas in this paper are just symbols.**
> >
> > We use mathematical symbols to succinctly write various evaluation metric ( coverage, hybrid accuracy, budget ) such that the final optimization problem can be written in a clear and concise manner. This allows to write the exactly constrained optimization problem that represents the problem at hand. For example,
> > - Eq.(3) represents the optimization problem that includes architecture search as well as training a hybrid model
> > - Eq.(4) represents the inner optimization present in Eq.(3), i.e., given any fixed architectures (base, global), how do we obtain hybrid models that follow a given budget constraint.
> > - Eq.(5) represents the router learning problem given pre-trained base and global models
> >
> > It is unclear why the reviewer thinks writing these optimization problems in words is better than succinctly writing them in mathematical form that can be relaxed with loss functions, yielding an algorithmic procedure to solve the learning tasks (precisely the Algorithm 1).

---

> > > ### Author Response · Authors · 2022-11-16
> > > **Clarity 2.2, 2.4: Moreover, the so-called "coverage constraint promotes " above Eqn (6) makes sense for efficiency improvement that reduces communication to the cloud. However, it is unreasonable from the optimization perspective since constraints tell only feasibility. The statements on page 6 before the "Focusing competency and loss functions" paragraph are not convincing. A few examples for illustration will be better.**
> > >
> > > Thank you for bringing this up, since this idea is crucial to the method. Let us clarify. For a fixed dataset $(x_i, y_i)_{i = 1}^n,$ the oracle $o$ is an empirical optimum to an empirical version of (Eq.5) for any value of $C_\varrho \ge C^* := \frac{1}{n} \sum \mathbf{1} (b(x_i) \neq g(x_i) = y_i)$. In particular, $C^*$ is the smallest value of $C_\varrho$ for which the empirical version of (Eq.5) has objective value $0$, and $o$ is the unique function that is optimal for all $C_\varrho \ge C^*$ - for this reason, we view $o$ as an ``ideal'' router.
> > >
> > >
> > > If $C_\varrho < C^*,$ then different $x_i$ s for which $b(x_i) \neq g(x_i) = y_i$ are not differentiated by the empirical optimisation problem - any function that assigns $n(C^* - C_\varrho)$ of these points to the base is optimal (and suffers an empirical accuracy loss of $C^* - C_\varrho$ relative to $o$). However, some of these assignments will lead to better router models upon training than others. As such, in order to meet the coverage constraint, we must introduce an inductive bias to decide which of these points should be assigned to base. Ideally, this choice of assignment should promote some regularity in the supervision, since this would lead to a better chance of fitting these labels with a small router, and thus better generalisation of the router. Unfortunately this is a very ill posed goal, and it is unclear how to exactly achieve this. Our approach is to exploit a penalised binary classification scheme for this - we introduce $o(x)$ as a superivision for the router, and then indirectly impose an inductive bias by introducing a penalty term in the loss to enforce that the empirical coverage is close to the target. The thought here is that rather than trying to design some crude assignment rule, we are better off exploiting existing algorithms that already deal with noisy data to adapt to the regularity present in the data.

---

> > > > ### Author Response · Authors · 2022-11-16
> > > > **Clarity 2.3: The statement around Eqn (7) is confusing. The "cov" term even seems unrelated. The "converge" is an upper bound of how many samples need to be delivered to the cloud to achieve a communication efficiency requirement. The authors should explicitly state such a relationship in words.**
> > > >
> > > > Thank you for pointing this out. We will update our wording to clarify this. Below, we provide the updated explanation.
> > > >
> > > > The term
> > > > $$\max(0, \textrm{cov} - (1 -   \frac{1}{N} \sum_x ( r_1(x) - r_0(x) )_{+} ) )) $$
> > > >
> > > > is equivalent to the following term
> > > >
> > > > $$\max(0,  \frac{1}{N} \sum_x ( r_1(x) - r_0(x) )_{+} ) - (1 - \textrm{cov}) )$$
> > > >
> > > > where  $\textrm{cov}$ is the fraction of examples inferred by the base model.
> > > >
> > > > As a result, $1-\textrm{cov}$ is the fraction of example, we need to route to the cloud. $\frac{1}{N} \sum_x ( r_1(x) - r_0(x))_+$ quantity estimates the fraction of examples sent to the global model by smoothening the indicator loss $\frac{1}{N} \sum_x \mathbf{1}( r_1(x) > r_0(x) )$. Above written term penalizes the router whenever the estimated fraction of examples sent to the global model is larger than the required $1-\textrm{cov}$ constraint.

---

### Official Review · Reviewer_q8G9 · 2022-10-25

**Confidence:** 4
**Correctness:** 4
**Technical Novelty And Significance:** 3
**Empirical Novelty And Significance:** 4
**Recommendation:** 8

**Clarity, Quality, Novelty And Reproducibility:**

The paper is well-written and very detailed. The problem statement is solid, and the method is unique. There may be some missing details, but I hope the code will be released soon.

**Details Of Ethics Concerns:**

The authors addressed potential considerations in the appendix. There seems to be no other ethical concern to this paper.

**Strength And Weaknesses:**

Strengths:
- The proposed framework works end-to-end, including the routing module. Especially, the routing module is trained via proxy supervision (using oracle), which is a clever way to incorporate all modules in the training.
- Experiments are carefully designed and performed. More importantly, actual MCU systems are used for the evaluation. “Hybrid accuracy” seems to be a good metric for these two-stage inference systems.
- Previous approaches (split-computation, dynamic network, early exit, etc.) are thoroughly studied and compared.

Weaknesses:
- Minor: what if the global model fails to predict correctly? It seems that the overall process assumes the global model is always better than the local model.
- Just a question; doesn’t the communication cost dependent on the model size, resolutions, etc.? It seems that such cost is somewhat fixed through the paper.

**Summary Of The Paper:**

The paper proposes an advanced two-stage inference strategy to reduce resource usage and communication cost. The key idea is that 1) use different edge and cloud models, 2) jointly train the router module, and 3) control the cost during training, dependent on the target environment. As a result, the proposed method achieves near-upper-bound performance in diverse resource constraints.

**Summary Of The Review:**

Overall, the paper tries to solve an important and practical issue. The proposed method is novel, and the experimental results support the claim sufficiently. I believe this paper is helpful for both research and production areas.

---

> ### Author Response · Authors · 2022-11-16
> **Communication cost depends on model size, resolutions, etc.**
>
> We agree that communication cost depends on many factors. This is precisely why we introduce the coverage metric ( fraction of examples inferred on the base model ), and we use it extensively in most of our empirical evaluations (Table 4, 5) for a fair comparison against competitive dynamic neural network baseline such as entropy thresholding. Coverage is agnostic to the input characteristics such as resolution that decide the transmission time. Note that in our setup the communication cost does not depend on the model size since we only transmit the input image. We discuss these aspects in Sec.2 (Modelling Resource Usage). In addition, we model different communication regimes in Figure 2.

---

> > ### Comment · Reviewer_q8G9 · 2022-11-19
> > **Thank you for the authors' response**
> >
> > I appreciate your response. The answers make sense to me, and I believe the revised manuscript addresses many potential concerns. Thank you for your hard work.

---

> ### Author Response · Authors · 2022-11-16
> **What if the global model fails to predict correctly?**
>
> Our oracle routing supervision only allocates the example to the global model whenever the global model is correct. As discussed in Sec.2.1 (Learning Routers via Proxy Supervision), the oracle routes different examples as follows:
>
> - Whenever the base is correct, oracle keeps the example on the base.
> - Whenever base is incorrect and global is correct, it sends the example to global.
> - Whenever base and global both are incorrect, it keeps the example on the base. ( This reduces communication to the cloud where the global model is going to be incorrect anyways. This is a very possible scenario in the Imagenet dataset. For ex., in our Figure 2 and Table 3,4,5, the global accuracy is only $80$%.)
>
> The assumption that the global model is always better than the local model should be natural since there is a large capacity difference between the global and local model (for ex., as seen from Table 1, global model has $9.1$MB parameter storage while base model has $0.6$MB parameter storage ). But note that this assumption does not affect the problem formulation as the oracle supervision always prefers the base model whenever base is correct irrespective of the global model correctness.

---

### Official Review · Reviewer_TiF9 · 2022-10-27

**Confidence:** 2
**Clarity, Quality, Novelty And Reproducibility:** The article is clear and looks reprod…
**Correctness:** 4
**Technical Novelty And Significance:** 2
**Empirical Novelty And Significance:** 3
**Recommendation:** 6

**Details Of Ethics Concerns:**

No ethics concern

**Strength And Weaknesses:**

Strength:

1. The paper provide a clear illustration of their methods and especially the comparison with the existing approaches.
2. The experiments are setup solidly and the covered settings and the metrics are comprehensive

Weakness:
1. According to Figure 2, improvements over dynamic methods may not be good enough to justify the significance
2. Limited novelty: alternating optimization to solve the coupled optimized parameters seem to be mentioned before by some works, such as [1] and [2]. And anyway it seems to be an intuitive approach.

[1] Auto-NBA: Efficient and Effective Search Over the Joint Space of Networks, Bitwidths, and Accelerators

[2] A3C-S: Automated Agent Accelerator Co-Search towards Efficient Deep Reinforcement Learning

**Summary Of The Paper:**

This paper proposed a novel hybrid design where an edge-device selectively queries the cloud only on those hard instances that the cloud can classify correctly to optimize accuracy under latency and edge-device constraints. An end-to-end method to train neural architectures, base predictors, and routing models is also proposed which is enabled by a novel proxy supervision for training routing models.

**Summary Of The Review:**

A clear article with simple yet relatively effective methods to solve the efficiency and accuracy dilemma of edge and cloud DNN execution. Would be great if the authors could provide more novelty and improvements justification.

---

> ### Author Response · Authors · 2022-11-16
> **Marginal Improvements Over Dynamic Methods.**
>
> **As such, the gains of hybrid scheme must be seen through the lens of how much latency can be gained for the same level of accuracy. On this important metric we do see significant gains.**
>
> Let us examine Table 3 and Fig. 2 closely. We can see that on the ImageNet dataset with the tiny MCU, to achieve the same accuracy of 74.7, the entropic methods (which include dynamic-nets) exhibit a latency of 1400ms while our method has a latency of 1000. Furthermore, our method is close to oracle performance. Next, notice that to obtain cloud accuracy of 79.9, the dynamic nets end up transmitting all inputs to the cloud, while we have a latency of 1400. Alternatively, if we do not want to use latency as a metric, we can use coverage and conclude that we transmit 25\% fewer examples to the cloud. As such these are very significant gains (25\%).
>
> As an aside, note that on the Imagenet dataset, at same computational budget, improvements of $1-2$% are considered significant (see MobileNetV3, EfficientNet works). Figure 2 shows that hybrid models achieve on-cloud accuracy at $30$% coverage while dynamic neural networks achieve significantly less accuracy at this coverage level. Similarly,  Table 4 shows that at $70$% coverage, hybrid models outperform the dynamic neural networks by up to $2$% increase in accuracy.

---

> ### Author Response · Authors · 2022-11-16
> **Limited Novelty**
>
> Please see https://openreview.net/forum?id=jpR98ZdIm2q&noteId=qPTCtbDaeiT
>
> In summary, we point out that our novelty lies in the proposed proxy oracle supervision for the router and the routing induced weighted losses for learning the base and global models. Such a setup is markedly different than the existing literature where router is learnt through indirect supervision available in the hybrid accuracy.

---

### Official Review · Reviewer_YaHL · 2022-10-31

**Confidence:** 3
**Correctness:** 3
**Technical Novelty And Significance:** 2
**Empirical Novelty And Significance:** 1
**Recommendation:** 3

**Clarity, Quality, Novelty And Reproducibility:**

The paper is written clearly. The contributions seem incremental from an ML perspective. From a Systems perspective, the optimization goal seems a bit unrealistic, as explained above.

**Strength And Weaknesses:**

Strengths:
- The paper is well-organized.
- Efficient inference on edge devices is a timely topic.
- The idea of not sending the data to the server if it's not going to bring any improvement sounds interesting.

Weaknesses:
- Using average latency as an inference system metric here seems a bit unnatural to me. Controlling for the "average" latency is equivalent to maintaining the average sample throughput of the system. In the case of controlling for throughput, the batching mechanisms become quite effective: collect a batch of samples at the edge for T seconds, compress them as much as possible (even jointly if they have temporal correlation), and upload them to the server to be efficiently batch-processed by the server-class GPU. However, given the evaluations, I think the paper is not focusing on this regime of designs. On the other hand, in latency-sensitive applications, we are actually concerned about the "tail" latencies instead of the average. In that case, the proposed approach cannot provide fine-grained control over the metric of interest (tail latency): you are either going over the network with a 2sec delay or processing locally. For example, if we need a sub-second latency guarantee, the system cannot go over the network at all, and if we go for a 2.5sec latency guarantee, the system will process everything on the remote server.
- How does this work relate to the Mixture of Experts literature? Can we think of r as the gating network and b and g as two experts with different sizes? How does your performance compare against training such an MoE but only running the expert that is hard-selected by the gating network?
- "We are the first to provably reduce router learning to binary classification and exploit it for end-to-end training based on novel proxy supervision of routing models." This contribution might be a little bit overstated. For example, have you seen the routing section in [1]?

[1] Riquelme, C., Puigcerver, J., Mustafa, B., Neumann, M., Jenatton, R., Susano Pinto, A., Keysers, D. and Houlsby, N., 2021. Scaling vision with sparse mixture of experts. Advances in Neural Information Processing Systems, 34, pp.8583-8595.

**Summary Of The Paper:**

This paper proposes using a shallow network to route a subset of samples from the edge device to a remote server. The router, edge, and server models are trained jointly but iteratively.

**Summary Of The Review:**

Given that the evaluations only support latency-sensitive applications, but the optimization goal is not aligned with latency-sensitive systems, I think the paper can be more interesting for the systems community if it addresses either of the issues.

---

> ### Author Response · Authors · 2022-11-15
> **Is our optimization goal not aligned with latency-sensitive systems? 2sec delay or processing locally; Has no sub-second control.**
>
> We believe that reviewer has **not** considered a very important use-case. Virtual personal assistants (VPA) are increasingly gaining traction among mobile and wearable device users (a multi-billion dollar market [a]). We introduce this use-case in the revised paper for reference. Let us outline our basic argument to demonstrate how our problem setting is natural and reviewers concerns *are not valid* in this case.
>
> **What are unique aspects of querying and VPAs in mobile/wearable applications?**
>
> 1. *Query in VPA setting*: Users interact with VPAs by posing a query. For instance, for Alexa (Amazon), a user might ask ("What is the time?"). VPAs must rapidly respond (low-latency) to the user query. Unusually long delay results in user friction, and can limit future user engagement.
>
> 2. *Querying vs. Batch-Data*: Users query at a random time, and the system needs to be responsive. It is simply not meaningful to batch queries, say in a day/week, and respond to the user. This is a fundamental reason why reviewer suggested batching is not applicable for VPAs.
>
> 3. *Query Diversity*. User understands that some of her queries are simple ("What is the time?") and others are hard ("Play music I would like from the 80s"). For the harder queries, the user is conditioned to expect that harder questions will take a bit longer.
>
> 4. *High-Dimensional Queries*. VPAs are increasingly admitting acoustic and image inputs, and such inputs are high-dimensional. These queries require processing on large-scale if we need to guarantee high-accuracy for every query.
>
> **Summary** We cannot batch data, and larger tail latency is acceptable and to be expected for harder queries. Thus the question of users demanding good response rate on-average and uniformly good accuracy on all queries is a natural problem.
>
> **What are the system-level constraints for mobile-based VPAs?**
> As we point out (this point is clarified in the revised version) in the paper, due to device miniaturization coupled with high query diversity, it is simply not possible to respond accurately on-device for every query. Consequently, a few of the queries must inevitably go to the cloud for deeper processing. Furthermore, a VPN provider (eg. Apple/Google/Amazon) would like to reduce server load and would desire more on-device processing.
>
> *Are there prior works that present hybrid schema?* The fact that we must be adaptive and share processing between device and on-cloud is not new and has been discussed extensively (see our references or [c] and [d]). In fact [c] has nearly 900 citations, which does suggests the importance of the setup and problem. We also point out [b] for context from industry.
>
> **Learning Objective** With the above discussion serving as the background, it is natural to ask - *how can you train a hybrid model such that the maximum number of queries are handled on-device (coverage), while ensuring that we do not suffer accuracy degradation from what is achievable by transmitting to the cloud all the time?* While we set this up as a maximum coverage under no accuracy-degradation constraint, our method generalizes and we can systematically characterize the entire coverage/accuracy frontier.
>
> *Coverage vs. Latency* There is a 1-1 correspondence between coverage and latency (see Sec. 2), and as such our method also provides frontiers for latency (or any other resource for that matter).
>
> **Reviewer Comment:** How can we get fine grained control?
> We do not need to, as there is a widespread diversity in queries across time. Many are handled locally and the hard ones do suffer larger latencies. Furthermore, note that our method is general and we can optimize for any level of latency (see Fig 2b). In the way we setup the problem we can also optimize accuracy for any coverage level, and since coverage has 1-1 correspondence with latency, our method allows for arbitrary latencies. If this latency turns out to be low, we do end up transmitting all queries to the cloud, and thus all the system-level specs are fully exploited by our method.
>
> **Compelling Results for VPAs.** We provide solutions that significantly reduce the average latency and energy costs, while not impacting tail latency, i.e. - the typical performance is significantly improved, while the worst case performance is left largely unaffected, all while maintaining high accuracy requirements. For example, using our lightweight base models, at a marginal accuracy cost ($\approx 0.4\%$) one can reduce average latencies by $25\%,$ while increasing max latency by \textcolor{red}{$<10\%$} (Table 3).
>
> [a] https://www.globenewswire.com/news-release/2015/12/17/796353/0/en/Intelligent-Virtual-Assistant-Market-Worth-3-07Bn-By-2020.html
>
> [b] http://www.interspeech2020.org/uploadfile/pdf/Wed-1-2-7.pdf
>
> [c] https://ieeexplore.ieee.org/document/9586176
>
> [d] https://dl.acm.org/doi/10.1145/3093336.3037698

---

> > ### Author Response · Authors · 2022-11-16
> > **How does this work relate to MoE for instance, "Scaling vision with sparse mixture of experts, NeuRIPS 21"?**
> >
> > While the two problems are related, they are fundamentally different. We encounter a constrained learning problem (see Eq. 4 and 5 in paper and discussion above), while MoEs perform load-balancing among experts.
> >
> > **Load Balancing vs. Maximizing Coverage** MoE methods typically route between experts of similar capacity, and therefore focus a lot on load-balancing aspects. In contrast, our setup has a large capacity difference between the edge and global models, and further, ideal routing still needs to prefer the edge model over the cloud model to maintain high coverage. This difference in scenario renders most MoE methods inapplicable in our setting. Let us drill down a bit more to make this point clearer and show that the MoE method as outlined is fundamentally inapplicable.
> >
> > Consider the loss functions utilized in the MoE setting ([a]). There are three loss functions: the classification loss, which is standard, and a novel auxilliary loss consisting of importance loss, $L_{imp}$, and load loss $L_{load}$. The auxilliary loss are symmetric loss functions across experts, and penalize unequal expert load. To understand why using this loss is fundamentally flawed for our coverage maximization context, let us unpack them:
> > $$
> > L_{imp}(X) = (\frac{std(imp(X)}{mean(imp(X))})^2,\qquad L_{load}(X) = (\frac{std(load(X)}{mean(load(X))})^2
> > $$
> > In our context let us follow reviewer suggestion of viewing our setting as a two-expert problem; $imp(X)$ has two components composed of (soft) measures on how many times on the training dataset $X$, the base model, $b$, and cloud model, $g$, are utilized. Load is not of relevance to our setting since it applies to situations with three or more experts (see [a]). At a fundamental level, penalizing $L_{imp}(X)$ amounts to forcing a uniform distribution among expert usage. Applying this to our setting faces the following obstacle:
> >
> > **Need for Asymmetric Loss.** We typically want base to be more active (higher coverage). As such this means that we need an asymmetric loss (with more weight on cloud to reduce usage). However, both importance and load losses are symmetric. While one could re-weight $imp(X)$ and $load(X)$ to introduce asymmetry, this is a heuristic involving additional hyperparameters. Alternatively, we could set a lower penalty on the symmetric loss less, so that the algorithm is not penalized strongly for unbalanced loads. However, this also does not make sense because the symmetry between global and local load means that the router would prefer global (due to its higher accuracy).
> >
> > **Can we adapt MoE for our setting?** Yes. Such variants have been proposed and documented in our paper. In particular, MoE for budgeted settings have been proposed, e.g., "Adaptive Classification for Prediction Under a Budget" and "AppealNet: An Efficient and Highly-Accurate Edge/Cloud Collaborative Architecture for DNN Inference". The main difference here is that in contrast to the symmetric load and importance loss functions that encourage equal load among experts, this paper proposes an asymmetric loss by penalizing log of the probability of usage of the cloud model. While this makes sense, its performance is dominated by our method as we point out below.
> >
> > **Empirically our Method Dominates MoE variants** We discuss these methods in Page.4 (Dynamic Neural Network) and Appendix A.7. These methods directly optimize the hybrid accuracy under a budget constraint. As discussed in "Baseline Algorithms" in Sec.3, simple method such as entropy thresholding the edge device predictions outperforms these methods. This is again evident from the CIFAR-100 experiment shown in the Table~10 where entropy thresholding outperforms AppealNet.
> >
> > Our empirical evaluations (Figure 2 and Table 3,4,5) show that the proposed method convincingly outperforms these methods where routing is indirectly learnt by minimizing the hybrid accuracy. Our router is learnt using binary supervision through proxy oracle that promotes routing the examples to cloud whenever cloud is correct and the edge is incorrect.
> >
> > [a] Riquelme, C., Puigcerver, J., Mustafa, B., Neumann, M., Jenatton, R., Susano Pinto, A., Keysers, D. and Houlsby, N., 2021. Scaling vision with sparse mixture of experts. Advances in Neural Information Processing Systems, 34, pp.8583-8595.

---

> > > ### Author Response · Authors · 2022-11-16
> > > **What is the novelty?**
> > >
> > > It is apparent that our method dominates prior works. These include entropy thresholding, variants of MoE and methods based on dynamic neural network, and others. Let us attempt to answer why this is so below (although much of what we write below appears in the main paper and the appendix).
> > >
> > > **Summary.** If we pose the problem in terms of indicator losses, it follows that the router (as a function of base and cloud) reduces exactly to a oracle supervision problem. In particular, our analysis below will show that the router can pretend that its labels are $o(x)=\mathbf{1}_{b(x)\neq g(x)}$ and train using $\{(x,o(x))}$ as the dataset under a constraint on coverage. Thus, in summary we have
> > > 1. *Stronger Surrogate.* Our method exploits a stronger surrogate for loss for the router since it is based on the indicator loss, which is closer to the actual problem.
> > > 2. *Supervised Classification.* Router learning can be explicitly posed as a **supervised** binary classification problem (see argument below) for a given base, $b$, and cloud, $g$. As such the router has crisp discrete labels to train on.
> > > 3.  In contrast, MoE and other variants do not have explicit/discrete labels, and implicitly due to randomness in b, g evolution during training, the router does not have a clear signal on whether to pick $b$ or $g$ in any round. We see this in our experiments where the router has difficulty converging with the MoE variants.
> > >
> > > **Simplified analysis showing that learning a router can be reduced to supervised binary classification problem**
> > > Fundamentally, let us examine our problem through *indicator losses* and see how this translates to proposed binary classification. As such indicator losses are what we really wish to solve,
> > >
> > > $\min_{b,g} \min_r \mathbb{E} \left [(1-r(x))\mathbf{1}[b(x)\neq y] + r(x)\mathbf{1}[g(x)\neq y] \right ] \quad \mbox{s.t.} \quad \mathbb{E} r(x) \leq C$
> > >
> > > This is a simplified version of Eq. 4 in the paper with losses rather than accuracy reflected there. $r(x)$ is a binary variable that outputs zero if the input $x$ is not transmitted to the router. Let us consider the router's loss problem as a function of  $b$ and $g$. This parallels Eq. 5 but with loss instead of accuracy.
> > > $\min_r \mathbb{E} \left [(1-r(x))\mathbf{1}[b(x)\neq y] + r(x)\mathbf{1}[g(x)\neq y] \right ] \quad \mbox{s.t.} \quad \mathbb{E} r(x) \leq C$
> > >
> > > We see that, the problem is equivalent to:
> > > $\min_r \mathbb{E} \left [r(x)(\mathbf{1}[g(x)\neq y]-\mathbf{1}[b(x)\neq y])\right ]\quad \mbox{s.t.} \quad \mathbb{E} r(x) \leq C$
> > >
> > > It is not hard to check that $o(x) \triangleq \mathbf{1}[b(x)\neq g(x)] = (\mathbf{1}[b(x)\neq y]-\mathbf{1}[g(x)\neq y])$ under the natural assumption that when base is correct cloud is correct. (Note that in the paper we do not impose this assumption, and we also account for the *unlikely* situation of base being correct but cloud incorrect). We make this assumption here to simplify our analysis and show the reduction. It follows that our problem reduces to:
> > >
> > > $\min_r \mathbb{E}r(x)(1-o(x))\quad \mbox{s.t.} \quad \mathbb{E} r(x) \leq C$
> > >
> > > The objective says that we penalize transmits if there is agreement, and are agnostic otherwise. This is what is basically what appears in Sec. 2.1.
> > >
> > > To gain further intuition, we can further manipulate the expression and obtain a supervised binary classification problem. In particular, noting that for binary random variables, $r(x)(1-o(x)) = (1-o(x))\mathbf{1}[r(x)\neq o(x)]$, we obtain,
> > >
> > > $\min_r \mathbb{E}(1-o(x))\mathbf{1}[r(x)\neq o(x)]\quad \mbox{s.t.} \quad \mathbb{E} r(x) \leq C$
> > >
> > > The objective enforces penalty when there is an agreement and router chooses cloud, but for the case of disagreement the objective is agnostic. We can upperbound the objective to yield an unweighted problem: $\min_r \mathbb{E}\mathbf{1}[r(x)\neq o(x)]\quad \mbox{s.t.} \quad \mathbb{E} r(x) \leq C$
> > >
> > > We are now in a position to infer that the router must exactly solve a supervised binary classification problem with $o(x)$ serving as weights on the examples, and with the added constraint on coverage.

---

### Decision · Program_Chairs · 2023-01-20

**Decision:**

Accept: poster

**Justification For Why Not Higher Score:**

Reviewers liked the idea but were not very excited. The method proposed is sound but a very intuitive direction pursued by many prior works.

**Justification For Why Not Lower Score:**

The papers contribution is worthy of publication.

**Metareview: Summary, Strengths And Weaknesses:**

To improve edge efficiency and privacy the authors propose a hybrid inference approach wherein the inference is served either from the device or from the cloud depending on how confident the predictions are. Although the idea is quite natural and intuitive, the authors proposed a formal framework and develop sound global objectives and algorithms to make this a reality. Experiments are convincing and the proposal solves a very important practical problem.



**Note From Pc:**

if the above contains the word "oral" or "spotlight" please see: "oral" presentation means -> notable-top-5% and "spotlight" means -> notable-top-25%. As stated in our emails, we are disassociating presentation type from AC recommendations

**Summary Of Ac-Reviewer Meeting:**

All the reviewers except one are convinced that this paper is well written, solve a very important problem and the results are exciting. One reviewer had concerns about metric and existing methods. The authors did provide a reasonable rebuttal and in ACs opinions the concerns raised does not qualify for a grounds of rejections.

Overall, the paper is above bar for acceptance.